# Rapid learning of predictive maps with STDP and theta phase precession

**Tom M George[1†], William de Cothi[2†], Kimberly L Stachenfeld[3], Caswell Barry[2]***

[1]Sainsbury Wellcome Centre for Neural Circuits and Behaviour, University College London, London, United Kingdom; [2]Research Department of Cell and Developmental Biology, University College London, London, United Kingdom; [3]DeepMind, London, United Kingdom

***For correspondence:**
caswell.barry@ucl.ac.uk

[†]These authors contributed equally to this work

**Abstract** The predictive map hypothesis is a promising candidate principle for hippocampal function. A favoured formalisation of this hypothesis, called the successor representation, proposes that each place cell encodes the expected state occupancy of its target location in the near future. This predictive framework is supported by behavioural as well as electrophysiological evidence and has desirable consequences for both the generalisability and efficiency of reinforcement learning algorithms. However, it is unclear how the successor representation might be learnt in the brain. Error-driven temporal difference learning, commonly used to learn successor representations in artificial agents, is not known to be implemented in hippocampal networks. Instead, we demonstrate that spike-timing dependent plasticity (STDP), a form of Hebbian learning, acting on temporally compressed trajectories known as 'theta sweeps', is sufficient to rapidly learn a close approximation to the successor representation. The model is biologically plausible – it uses spiking neurons modulated by theta-band oscillations, diffuse and overlapping place cell-like state representations, and experimentally matched parameters. We show how this model maps onto known aspects of hippocampal circuitry and explains substantial variance in the temporal difference successor matrix, consequently giving rise to place cells that demonstrate experimentally observed successor representation-related phenomena including backwards expansion on a 1D track and elongation near walls in 2D. Finally, our model provides insight into the observed topographical ordering of place field sizes along the dorsal-ventral axis by showing this is necessary to prevent the detrimental mixing of larger place fields, which encode longer timescale successor representations, with more fine-grained predictions of spatial location.

## Editor's evaluation

This theoretical work is important in that it bridges neural mechanisms within the hippocampus with the abstract computations it is thought to support for reinforcement learning. The study offers a potential mechanism by which spike timing dependent plasticity and theta phase precession within spiking neurons in CA3 and CA1 can yield successor representations. The simulations are compelling in that they continue to hold even when some of the simple but less realistic assumptions are relaxed in support of more realistic scenarios consistent with biological data.

## Introduction

Knowing where you are and how to navigate in your environment is an everyday existential challenge for motile animals. In mammals, a key brain region supporting these functions is the hippocampus (*Scoville and Milner, 1957*; *Morris et al., 1982*), which represents self-location through the population activity of place cells – pyramidal neurons with spatially selective firing fields (*O'Keefe and*

*Dostrovsky, 1971*). Place cells, in conjunction with other spatially tuned neurons (*Taube et al., 1990*; *Hafting et al., 2005*), are widely held to constitute a 'cognitive map' encoding information about the relative location of remembered locations and providing a basis upon which to flexibly navigate (*Tolman, 1948*; *O'Keefe and Nadel, 1978*).

The hippocampal representation of space incorporates spike time and spike rate based encodings, with both components conveying broadly similar levels of information about self-location (*Skaggs et al., 1996b*; *Huxter et al., 2003*). Thus, the position of an animal in space can be accurately decoded from place cell firing rates (*Wilson and McNaughton, 1993*) as well as from the precise time of these spikes relative to the background 8–10 Hz theta oscillation in the hippocampal local field potential (*Huxter et al., 2003*). The latter is made possible since place cells have a tendency to spike progressively earlier in the theta cycle as the animal traverses the place field – a phenomenon known as phase precession (*O'Keefe and Recce, 1993*). Therefore, during a single cycle of theta the activity of the place cell population smoothly sweeps from representing the past to representing the future position of the animal (*Maurer et al., 2006*), and can simulate alternative possible futures across multiple cycles (*Johnson and Redish, 2007*).

In order for a cognitive map to support planning and flexible goal-directed navigation, it should incorporate information about the overall structure of space and the available routes between locations (*Tolman, 1948*; *O'Keefe and Nadel, 1978*). Theoretical work has identified the regular firing patterns of entorhinal grid cells with the former role, providing a spatial metric sufficient to support the calculation of navigational vectors (*Bush et al., 2015*; *Banino et al., 2018*). In contrast, associative place cell – place cell interactions have been repeatedly highlighted as a plausible mechanism for learning the available transitions in an environment (*Muller et al., 1991*; *Blum and Abbott, 1996*; *Mehta et al., 2000*). In the hippocampus, such associative learning has been shown to follow a spike-timing dependent plasticity (STDP) rule (*Bi and Poo, 1998*) – a form of Hebbian learning where the temporal ordering of spikes between presynaptic and postsynaptic neurons determines whether long-term potentiation or depression occurs. One of the consequences of phase precession is that correlates of behaviour, such as position in space, are compressed onto the timescale of a single theta cycle and thus coincide with the time-window of STDP $\mathcal{O}(20 - 50\,\mathrm{ms})$ (*Skaggs et al., 1996b*; *Mehta et al., 2000*; *Mehta, 2001*; *Mehta et al., 2002*). This combination of theta sweeps and STDP has been applied to model a wide range of sequence learning tasks (*Jensen and Lisman, 1996*; *Koene et al., 2003*; *Reifenstein et al., 2021*), and as such, potentially provides an efficient mechanism to learn from an animal's experience – forming associations between cells which are separated by behavioural timescales much larger than that of STDP.

Spatial navigation can readily be understood as a reinforcement learning problem – a framework which seeks to define how an agent should act to maximise future expected reward (*Sutton and Barto, 1998*). Conventionally, the value of a state is defined as the expected cumulative reward that can be obtained from that location with some temporal discount applied. Thus, the relationship between states and the rewards expected from those states are captured in a single value which can be used to direct reward-seeking behaviour. However, the computation of expected reward can be decomposed into two components – the successor representation, a predictive map capturing the expected location of the agent discounted into the future, and the expected reward associated with each state (*Dayan, 1993*). Such segregation yields several advantages since information about available transitions can be learnt independently of rewards and thus changes in the locations of rewards do not require the value of all states to be re-learnt. This recapitulates a number of long-standing theory of hippocampus which state that hippocampus provides spatial representations that are independent of the animal's particular goal and support goal-directed spatial navigation (*Redish and Touretzky, 1998*; *Burgess et al., 1997*; *Koene et al., 2003*; *Hasselmo and Eichenbaum, 2005*; *Erdem and Hasselmo, 2012*).

A growing body of empirical and theoretical evidence suggests that the hippocampal spatial code functions as a successor representations (*Stachenfeld et al., 2017*). Specifically, that the activity of hippocampal place cells encodes a predictive map over the locations the animal expects to occupy in the future. Notably, this framework accounts for phenomena such as the skewing of place fields due to stereotyped trajectories (*Mehta et al., 2000*), the reorganisation of place fields following a forced detour (*Alvernhe et al., 2011*), and the behaviour of humans and rodents whilst navigating physical, virtual, and conceptual spaces (*Momennejad et al., 2017*; *de Cothi et al., 2022*). However,

the successor representation is typically conceptualised as being learnt using the temporal difference learning rule (*Russek et al., 2017*; *de Cothi and Barry, 2020*), which uses the prediction error between expected and observed experience to improve the predictions. Whilst correlates of temporal difference learning have been observed in the striatum during reward-based learning (*Schultz et al., 1997*), it is less clear how it could be implemented in the hippocampus to learn a predictive map. In this context, we hypothesised that the predictive and compression properties of theta sweeps, combined with STDP in the hippocampus, might be sufficient to approximately learn a successor representation.

We simulated the synaptic weights learnt due to STDP between a set of synthetic spiking place cells and show they closely resemble the weights of a successor representation learnt with temporal difference learning. We found that the inclusion of theta sweeps with the STDP rule increased the efficiency and robustness of the learning, with the STDP weights being a close approximation to the temporal difference successor matrix. Further, we find no fine tuning of parameters is needed – biologically determined parameters are optimal to efficiently approximate a successor representation and replicate experimental results synonymous with the predictive map hypothesis, including the behaviourally biased skewing of place fields (*Mehta et al., 2000*; *Stachenfeld et al., 2017*) in realistic one- and two-dimensional environments. Finally, we use the simulation of STDP with theta sweeps to generate insight into the observed topographical ordering of place field sizes along the dorsal-ventral hippocampal axis (*Kjelstrup et al., 2008*), by observing that such organisation is necessary to prevent the detrimental mixing of larger place fields, which approximate longer timescale successor representations (*Momennejad and Howard, 2018*), with more fine-grained predictions of future spatial location. Our model, focussing on the role of theta sweeps and STDP in learning a hippocampal predictive

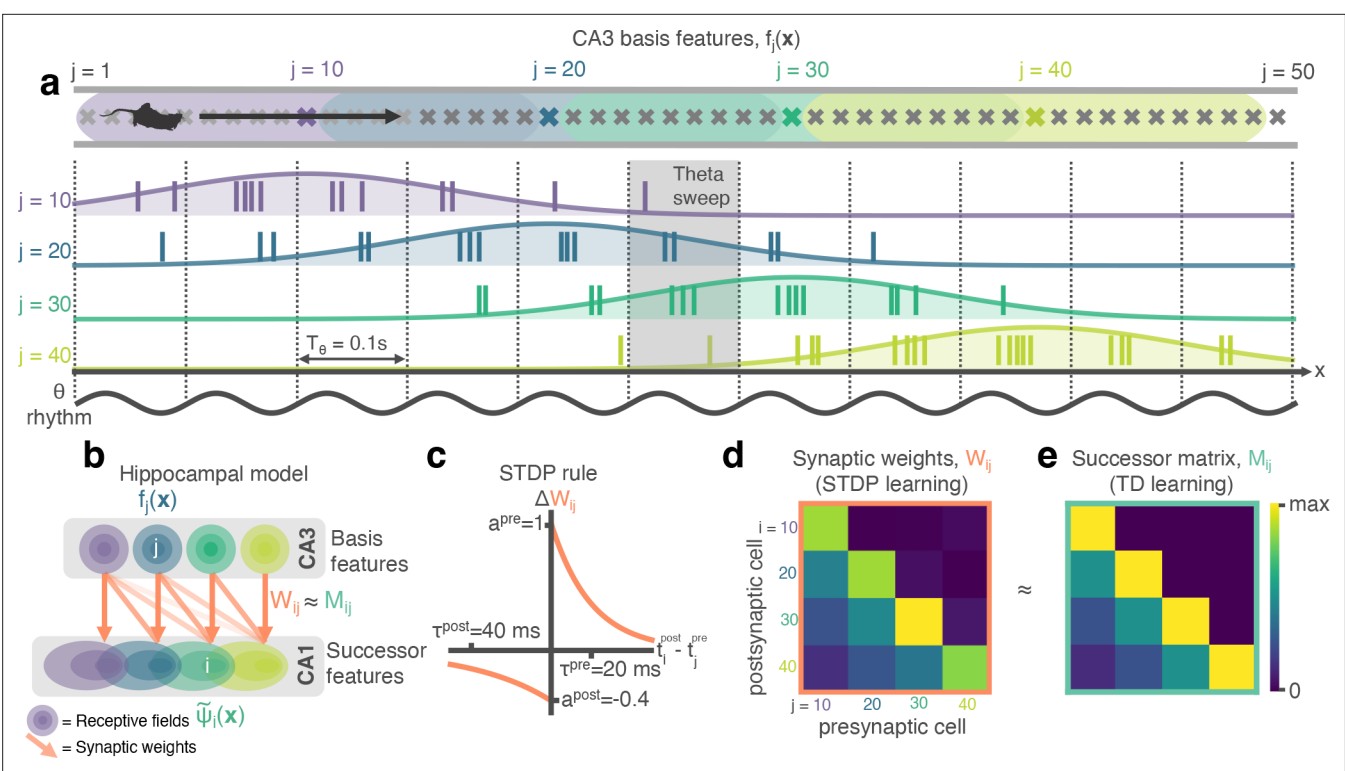

**Figure 1.** STDP between phase precessing place cells produces successor representation-like weight matrices. (**a**) Schematic of an animal running left-to-right along a track. 50 cells phase precess, generating theta sweeps (e.g. grey box) that compress spatial behaviour into theta timescales (10 Hz). (**b**) We simulate a population of CA3 'basis feature' place cells which linearly drive a population of CA1 'STDP successor feature' place cells through the synaptic weight matrix $W_{ij}$. (**c**) STDP learning rule; pre-before-post spike pairs ($t_i^{post} - t_j^{pre} > 0$) result in synaptic potentiation whereas post-before-pre pairs ($t_i^{post} - t_j^{pre} < 0$) result in depression. Depression is weaker than potentiation but with a longer time window, as observed experimentally. (**d**) Simplified schematic of the resulting synaptic weight matrix, $W_{ij}$. Each postsynaptic cell (row) fires just after, and therefore binds strongly to, presynaptic cells (columns) located to the left of it on the track. (**e**) Simplified schematic of the successor matrix (*Equation 3*) showing the synaptic weights after training with a temporal difference learning rule, where each CA1 cell converges to represent the successor feature of its upstream basis feature. Backwards skewing (successor features 'predict' upcoming activity of their basis feature) is reflected in the asymmetry of the matrix, where more activity is in the lower triangle, similar to panel d.

map, is part of a growing body of recent work emphasising hippocampally plausible mechanisms of learning successor representations, such as using hippocampal recurrence (*Fang et al., 2023*) or synaptic learning rules which bootstrap long-range predictive associations (*Bono et al., 2023*).

## Results

We set out to investigate whether a combination of STDP and phase precession is sufficient to generate a successor representation-like matrix of synaptic weights between place cells in CA3 and downstream CA1. The model comprises of an agent exploring a maze where its position $\mathbf{x}(t)$ is encoded by the instantaneous firing of a population of $N$ CA3 basis features, each with a spatial receptive field $f_j^x(\mathbf{x})$ given by a thresholded Gaussian of radius 1 m and 5 Hz peak firing rate. As the agent traverses the receptive field, its rate of spiking is subject to phase precession $f_j^\theta(\mathbf{x}, t)$ with respect to a 10 Hz theta oscillation. This is implemented by modulating the firing rate by an independent phase precession factor which varies according to the current theta phase and how far through the receptive field the agent has travelled (*Chadwick et al., 2015*) (see Methods and *Figure 1a*) such that, in total, the instantaneous firing rate of the $j^{\text{th}}$ basis features is given by:

$$f_j(\mathbf{x}, t) = f_j^x(\mathbf{x})f_j^\theta(\mathbf{x}, t). \tag{1}$$

CA3 basis features $f_j$ then linearly drive downstream CA1 'STDP successor features' $\tilde{\psi}_i$ (*Figure 1b*)

$$\tilde{\psi}_i(\mathbf{x}, t) = \sum_j \mathsf{W}_{ij} f_j(\mathbf{x}, t). \tag{2}$$

Using an inhomogeneous Poisson process, the firing rates of the basis and STDP successor features are converted into spike trains which cause learning in the weight matrix $\mathsf{W}_{ij}$ according to an STDP rule (see Methods and *Figure 1c*). The STDP synaptic weight matrix $\mathsf{W}_{ij}$ (*Figure 1d*) can then be directly compared to the temporal difference (TD) successor matrix $\mathsf{M}_{ij}$ (*Figure 1e*), learnt via TD learning on the CA3 basis features (the full learning rule is derived in Methods and shown in *Equation 27*). Further, the TD successor matrix $\mathsf{M}_{ij}$ can also be used to generate the 'TD successor features':

$$\psi_i(\mathbf{x}) = \sum_j \mathsf{M}_{ij} f_j^x(\mathbf{x}), \tag{3}$$

allowing for direct comparison and analyses with the STDP successor features $\tilde{\psi}_i$ (*Equation 2*), using the same underlying firing rates driving the TD learning to sample spikes for the STDP learning. This abstraction of biological detail avoids the challenges and complexities of implementing a fully spiking network, although an avenue for correcting this would be the approach of *Brea et al., 2016* and *Bono et al., 2023*. In our model phase, precession generates theta sweeps (*Figure 1a*, grey box) as cells successively visited along the current trajectory fire at progressively later times in each theta cycle. Theta sweeps take the current trajectory of the agent and effectively compress it in time. As we show below these compressed trajectories are important for learning successor features.

### The STDP learned synaptic weight matrix closely approximates the TD successor matrix

We first simulated an agent with $N = 50$ evenly spaced CA3 place cell basis features on a 5 m circular track (linear track with circular boundary conditions to form a closed loop, *Figure 2a*). The agent moved left-to-right at a constant velocity for 30 min, performing ~58 complete traversals of the loop. The STDP weights learnt between the phase precessing basis features and their downstream STDP successor features (*Figure 2b*) were markedly similar to the successor representation matrix generated using temporal difference learning applied to the same basis features under the same conditions (*Figure 2c*, element-wise Pearson correlation between matrices $R^2 = 0.87$). In particular, the agent's strong left-to-right behavioural bias led to the characteristic asymmetry in the STDP weights predicted by successor representation models (*Stachenfeld et al., 2017*), with both matrices dominated by a wide band of positive weight shifted left of the diagonal and negative weights shifted right.

To compare the structure of the STDP weight matrix $\mathsf{W}_{ij}$ and TD successor matrix $\mathsf{M}_{ij}$, we aligned each row on the diagonal and averaged across rows (see Methods), effectively calculating the mean distribution of learnt weights originating from each basis feature (*Figure 2d*). Both models exhibited

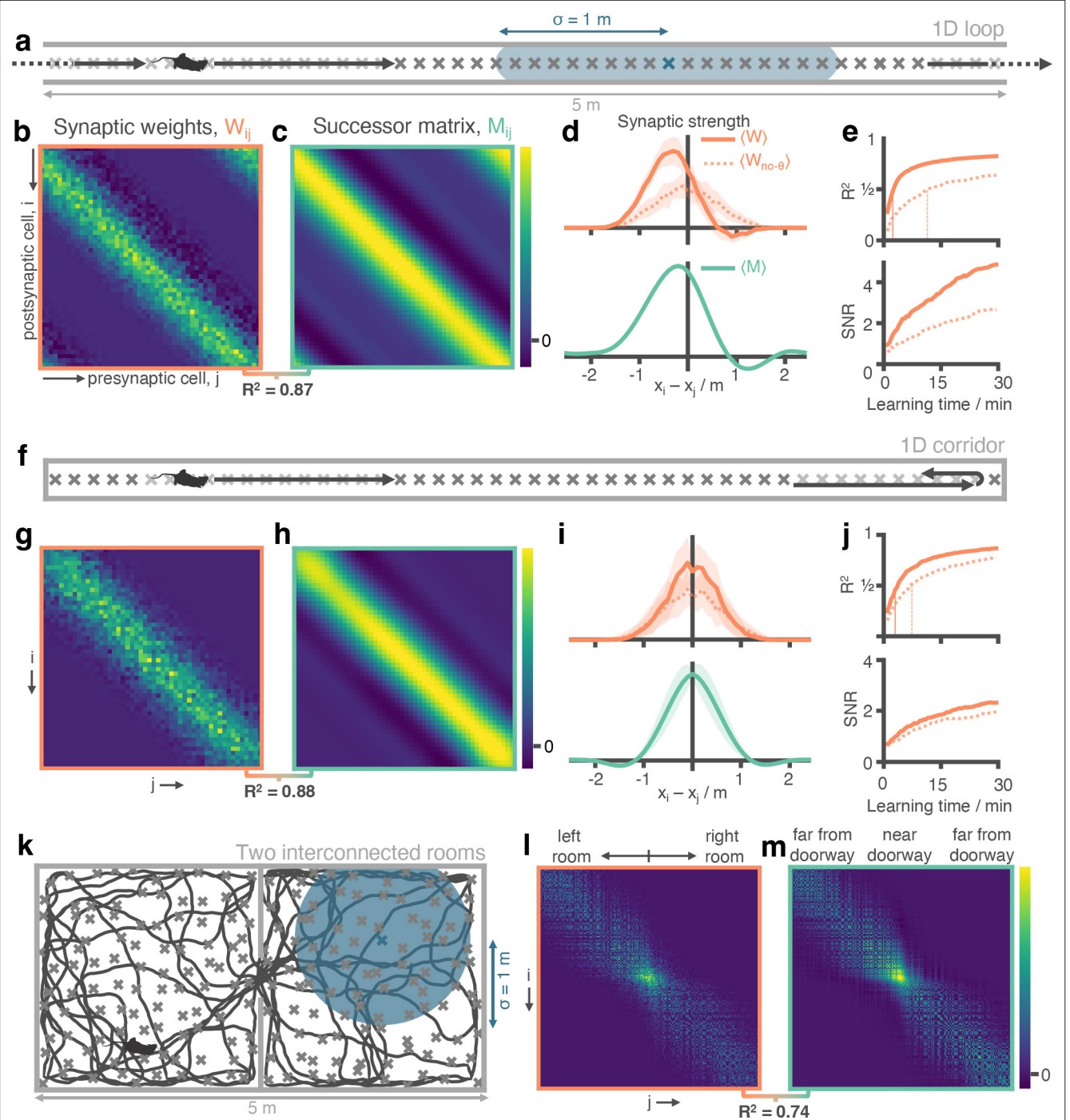

**Figure 2.** Successor matrices are rapidly approximated by STDP applied to spike trains of phase precessing place cells. (**a**) Agents traversed a 5 m circular track in one direction (left-to-right) with 50 evenly distributed CA3 spatial basis features (example thresholded Gaussian place field shown in blue, radius $\sigma = 1$ m). (**b&c**) After 30 min, the synaptic weight matrix learnt between CA3 basis features and CA1 successor features strongly resembles the equivalent successor matrix computed by temporal difference learning. Rows correspond to CA1, columns to CA3. (**d**) To compare the distribution of weights, matrix rows were aligned on the diagonal and averaged over rows (mean ± standard deviation shown). (**e**) Against training time, we plot (top) the R2 between the synaptic weight matrix and successor matrix and (bottom) the signal-to-noise ratio of the synaptic matrix. Vertical lines show time where R2 reaches 0.5. (**f-j**) Same as panels a-e except the agent turns around at each end of the track. The average policy is now unbiased with respect to left and right, as can be seen in the diagonal symmetry of the matrices. (**k-m**) As in panels a-c except the agent explores a two dimensional maze where two rooms are joined by a doorway. The agent follows a random trajectory with momentum and is biased to traverse doorways and follow walls.

*Figure 2 continued on next page*

*Figure 2 continued*

The online version of this article includes the following figure supplement(s) for figure 2:

**Figure supplement 1.** STDP and phase precession combine to make a good approximation of the SR independent of place cell size and running speed statistics.

**Figure supplement 2.** The STDP and phase precession model learns predictive maps irrespective of the weight initialisation and the weight updating schedule.

**Figure supplement 3.** A hyperparameter sweep over STDP and phase precession parameters shows that biological parameters are suffice, and are near-optimal for approximating the successor features.

**Figure supplement 4.** Biological phase precession parameters are optimal for learning the SR.

a similar distribution, with values smoothly ramping up to a peak just left of centre, before a sharp drop-off to the right caused by the left-to-right bias in the agent's behaviour. In the network trained by TD learning this is because CA3 place cells to the left of (i.e. preceding) a given basis feature are reliable predictors of that basis feature's future activity, with those immediately preceding it being the strongest predictors and thus conferring the strongest weights to its successor feature. Conversely, the CA3 place cells immediately to the right of (i.e. after) this basis feature are the furthest they could possibly be from predicting its future activity, resulting in minimal weight contributions. Indeed, we observed some of these weights even becoming negative (*Figure 2d*) – necessary to approximate the sharp drop-off in predictability using the smooth Gaussian basis features. With the STDP model, the similar distribution of weights is caused by the asymmetry in the STDP learning rule combined with the consistent temporal ordering of spikes in a theta sweep. Hence, the sequence of spikes emitted by different cells within a theta cycle directly reflects the order in which their spatial fields are encountered, resulting in commensurate changes to the weight matrix. So, for example, if a postsynaptic neuron reliably precedes its presynaptic cell on the track, the corresponding weight will be reduced, potentially becoming negative. We note that weights changing their sign is not biologically plausible, as it is a violation of Dale's Law (*Dale, 1935*). This could perhaps be corrected with the addition of global excitation or by recruiting inhibitory interneurons.

Notably, the temporal compression afforded by theta phase precession, which brings behavioural effects into the millisecond domain of STDP, is an essential element of this process (*Lisman and Grace, 2005*; *Koene et al., 2003*). When phase precession was removed from the STDP model, the resulting weights failed to capture the expected behavioural bias and thus did not resemble the successor matrix – evidenced by the lack of asymmetry (*Figure 2d*, dashed line; ratio of mass either side of y-axis 4.54 with phase precession vs. 0.99 without) and a decrease in the explained variance of the TD successor matrix (*Figure 2e*, $R^2 = 0.87 \pm 0.01$ vs $R^2 = 0.63 \pm 0.02$ without phase precession). Similarly, without the precise ordering of spikes, the learnt weight matrix was less regular, having increased levels of noise, and converged over $4.5\times$ more slowly (*Figure 2e*; time to reach $R^2 = 0.5$: 2.5 vs 11.5 min without phase precession), still yet to fully converge over the course of 1 hr (*Figure 2— figure supplement 1a*). Thus, the ability to approximate TD learning appears specific to the combination of STDP and phase precession. Indeed, there are deep theoretical connections linking the two – see Methods section 5.9 for a theoretical investigation into the connections between TD learning and STDP learning augmented with phase precession. This effect is robust to variations in running speed (*Figure 2—figure supplement 1b*) and field sizes (*Figure 2—figure supplement 1c*), as well as scenarios where target CA1 cells have multiple firing fields (*Figure 2—figure supplement 2a*) that are updated online during learning (*Figure 2—figure supplement 2b–d*), or fully driven by spikes in CA3 (*Figure 2—figure supplement 2e*); see Methods for more details.

We also conducted a hyperparameter sweep to test if these results were robust to changes in the phase precession and STDP learning rule parameters (*Figure 2—figure supplement 3*). The sweep range for each parameter contained and extended beyond the 'biologically plausible' values used in this paper (*Figure 2—figure supplement 3a*). We found that optimised parameters (those which result in the highest final similarity between STDP and TD weight matrices, $W_{ij}$ and $M_{ij}$) were very close to the biological parameters already selected for our model from a literature search (*Figure 2—figure supplement 3 c,d* parameter references also listed in figure) and, when they were used, no drastic improvement was seen in the similarity between $W_{ij}$ and $M_{ij}$. The only exception was firing rate for which performance monotonically improved as it increased - something the brain likely cannot achieve

due to energy constraints. In particular, the parameters controlling phase precession in the CA3 basis features (*Figure 2—figure supplement 4a*) can affect the CA1 STDP successor features learnt, with 'weak' phase precession resembling learning in the absence of theta modulation (*Figure 2—figure supplement 4b,c*), biologically plausible values providing the best match to the TD successor features (*Figure 2—figure supplement 4d*) and 'exaggerated' phase precession actually hindering learning (*Figure 2—figure supplement 4e*; see methods for more details). Additionally, we find these CA1 cells go on to inherit phase precession from the CA3 population even after learning when they are driven by multiple CA3 fields (*Figure 2—figure supplement 4f*), and that this learning is robust to realistic phase offsets between the populations of CA3 and CA1 place cells (*Figure 2—figure supplement 4g*).

Next, we examined the correspondence between our model and the TD-trained successor representation in a situation without a strong behavioural bias. Thus, we reran the simulation on the linear track without the circular boundary conditions so the agent turned and continued in the opposite direction whenever it reached each end of the track (*Figure 2f*). Again, the STDP and TD successor representation weight matrices where remarkably similar ($R^2 = 0.88$; *Figure 2gh*) both being characterised by a wide band of positive weight centred on the diagonal (*Figure 2i*) – reflecting the directionally unbiased behaviour of the agent. In this unbiased regime, theta sweeps were less important though still confered a modest shape, learning speed, and signal-strength advantage over the non-phase precessing model (*Figure 2j*) – evidenced as an increased amount of explained variance ($R^2 = 0.88 \pm 0.01$ vs. $R^2 = 0.76 \pm 0.02$) and faster convergence (time to reach $R^2 = 0.5$; 3 vs 7.5 minutes).

To test if the STDP model's ability to capture the successor matrix would scale up to open field spaces, we implemented a 2D model of phase precession (see Methods) where the phase of spiking is sampled according to the distance travelled through the place field along the chord currently being traversed (*Jeewajee et al., 2014*). We then simulated both the agent in an environment consisting of two interconnected 2.5 × 2.5 m square rooms (*Figure 2k*) using an adapted policy modelling rodent foraging behaviour that is biased towards traversing doorways and following walls (*Raudies and Hasselmo, 2012*; see Methods and 10 minute sample trajectory shown in *Figure 2k*). After training for 2 hr of exploration, we found that the combination of STDP and phase precession was able to successfully capture the structure in the TD successor matrix (*Figure 2l–m*, $R^2 = 0.74$, TD successor matrix calculated over the same 2 hr trajectory).

## Theta sequenced STDP place cells show behaviourally biased skewing, a hallmark of successor representations

We next wanted to investigate how the similarities in weights between the STDP and TD successor representation models are conveyed in the downstream CA1 successor features. One hallmark of the successor representation is that strong biases in behaviour (for example, travelling one way round a circular track) induce a reliable predictability of upcoming future locations, which in turn causes a backward skewing in the resulting successor features (*Stachenfeld et al., 2017*). Such skewing, opposite to the direction of travel, has also been observed in hippocampal place cells (*Mehta et al., 2000*). Under strongly biased behaviour on the circular linear track, the biologically plausible STDP CA1 successor features (*Equation 2*) had a very high correlation with the TD successor features (*Equation 3*) predicted by successor theory (*Figure 3a*; $R^2 = 0.98 \pm 0.01$). Both exhibited a pronounced backward skew, opposite to the direction of travel (mean TD vs. STDP successor feature skewness: $= -0.39 \pm 0.01$ vs. $= -0.24 \pm 0.07$). Furthermore, both the STDP and TD successor representation models predict that such biased behaviour should induce a backwards shift in the location of place field peaks (*Figure 3a* left panel; TD vs. STDP successor feature shift in metres: $-0.28 \pm 0.00$ vs $-0.38 \pm 0.03$) – this phenomenon is also observed in the hippocampal place cells (*Mehta et al., 2000*), and our model accounts for the observation that more shifting and skewing is observed in CA1 place cells than CA3 place cells (*Dong et al., 2021*). As expected, when theta phase precession was removed from the model no significant skew or shift was observed in the STDP successor features. Similarly, the skew in field shape and shift in field peak were not present when the behavioural bias was removed (*Figure 3b*) – in this unbiased scenario, the advantage of the STDP model with theta phase precession was modest relative to the same model without phase precession ($R^2 = 0.99 \pm 0.01$ vs. $R^2 = 0.96 \pm 0.01$).

Examining the activity of CA1 cells in the two-room open field environment, we found an increase in the eccentricity of fields close to the walls (*Figure 3c & d*; average eccentricity of STDP successor

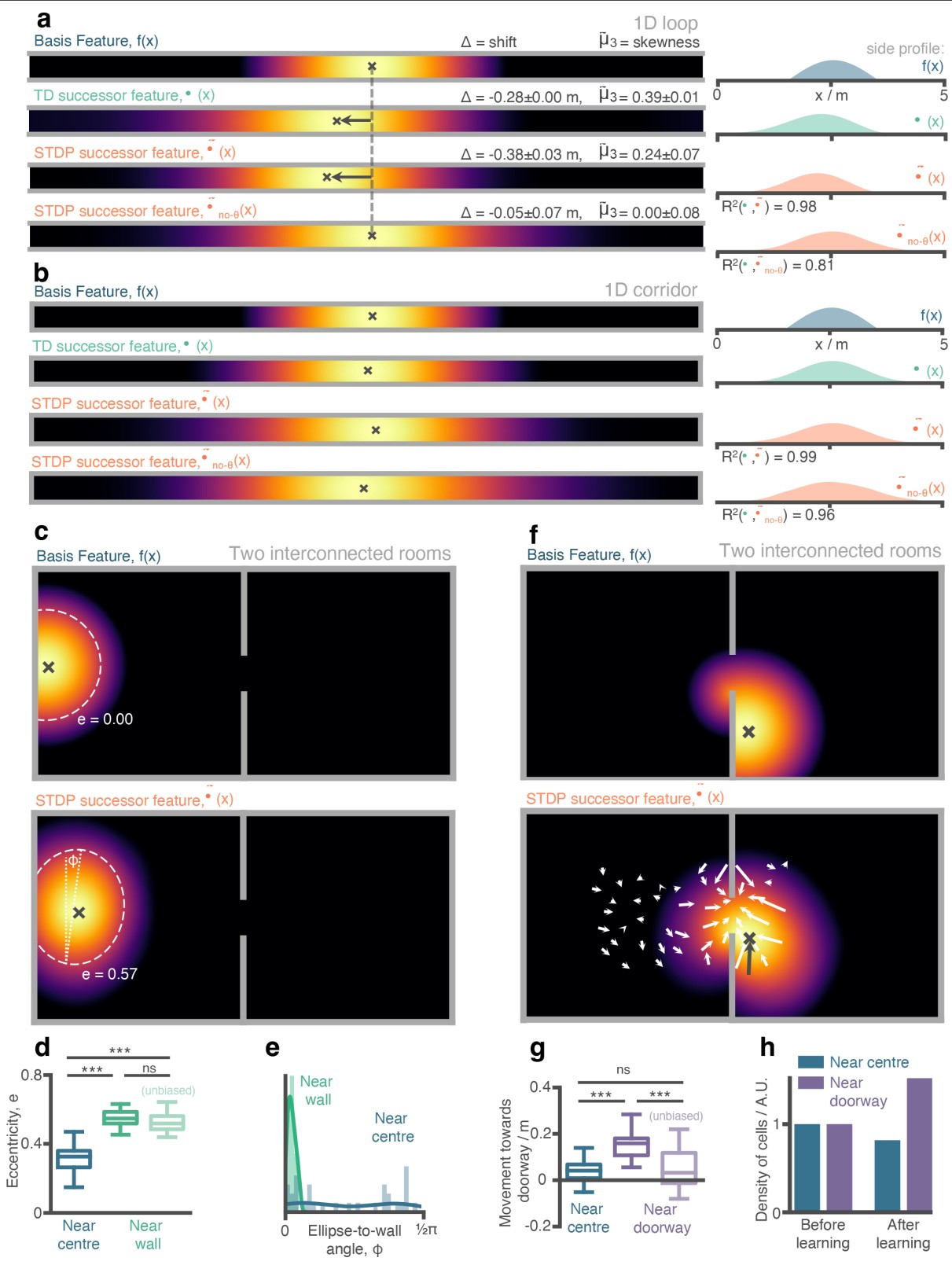

**Figure 3.** Place cells (aka. successor features) in our STDP model show behaviourally biased skewing resembling experimental observations and successor representation predictions. (**a**) In the loop maze (motion left-to-right), STDP place cells skew and shift backwards, and strongly resemble place cells obtained via temporal difference learning. This is not the case when theta phase precession is absent. (**b**) In the corridor maze, where travel in either direction is equally likely, place fields defuse in both directions due to the unbiased movement policy. (**c**) In the 2D maze, place cells (of geodesic

*Figure 3 continued on next page*

*Figure 3 continued*

Gaussian basis features) near the wall elongate along the wall axis (dashed line shows best fitting ellipse, angle construct show the ellipse-to-wall angle). (**d**) Place cells near walls have higher elliptical eccentricity than those near the centre of the environments. This increase remains even when the movement policy bias to follow walls is absent. (**e**) The eccentricity for fields near the walls is facilitated by an increase in the length of the place field along an axis parallel to the wall ($\phi$ close to zero). (**f**) Place cells near the doorway cluster towards it and expand through the doorway relative to their parent basis features. (**g**) The shift of place fields near the doorway towards the doorway is significant relative to place fields near the centre and disappears when the behavioural bias to cross doorways is absent. (**h**) The shift of place fields towards the doorway manifests as an increase in density of cells near the doorway after exploration.

features near vs. far from wall: $0.57 \pm 0.06$ vs. $0.33 \pm 0.07$). In particular, this increased eccentricity is facilitated by a shorter field width along the axis perpendicular to the wall (*Figure 3e*), an effect observed experimentally in rodent place cells (*Tanni et al., 2021*). This increased eccentricity of cells near the wall remained when the behavioural bias to follow walls was removed (*Figure 3d*; average eccentricity with vs. without wall bias: $0.57 \pm 0.06$ vs. $0.54 \pm 0.06$), thus indicating it is primarily caused by the inherent bias imposed on behaviour by extended walls rather than an explicit policy bias. Note that our ellipse fitting algorithm accounts for portions of the field that have been cut off by environmental boundaries (see methods & *Figure 3c*), and so this effect is not simply a product of basis features being occluded by walls.

In a similar fashion, the bias in the motion model we used - which is predisposed to move between the two rooms – resulted in a shift in STDP successor feature peaks towards the doorway (*Figure 3f & g*; inwards shift in metres for STDP successor features near vs. far from doorway: $0.15 \pm 0.06$ vs. $0.04 \pm 0.05$; with doorway bias turned off: $0.05 \pm 0.08$ vs. $0.04 \pm 0.05$). At the level of individual cells, this was visible as an increased propensity for fields to extend into the neighbouring room after learning (*Figure 3h*). Hence, although basis features were initialised as two approximately non-overlapping populations – with only a small proportion of cells near the doorway extending into the neighbouring room – after learning many cells bind to those on the other side of the doorway, causing their place fields to diffuse through the doorway and into to the other room (*Figure 3f*). This shift could partially explain why place cell activity is found to cluster around doorways (*Spiers et al., 2015*) and rewarded locations (*Dupret et al., 2010*) in electrophysiological experiments. Equally it is plausible that a similar effect might underlie experimental observations that neural representations in multi-compartment environments typically begin heavily fragmented by boundaries and walls but, over time, adapt to form a smooth global representations (e.g. as observed in grid cells by *Carpenter et al., 2015*).

## Multiscale successor representations are stored along the hippocampal dorsal-ventral axis by populations of differently sized place cells

Finally, we wanted to investigate whether the STDP learning rule was able form successor representation-like connections between basis features of different scales. Recent experimental work has highlighted that place fields form a multiscale representation of space, which is particularly noticeable in larger environments (*Tanni et al., 2021*; *Eliav et al., 2021*), such as the one modelled here. Such multiscale spatial representations have been hypothesised to act as a substrate for learning successor features with different time horizons – large-scale place fields are able to make predictions of future location across longer time horizons, whereas place cells with smaller fields are better placed to make temporally fine-grained predictions. Agents could use such a set of multiscale successor features to plan actions at different levels of temporal abstraction, or predict precisely which states they are likely to encounter soon (*Momennejad and Howard, 2018*). Despite this, what is not known is whether different sized place fields will form associations when subject to STDP coordinated by phase precession and what effect this would have on the resulting successor features. Hypothetically, consider a small basis feature cell with a receptive field entirely encompassed by that of a larger basis cell with no theta phase offset between the entry points of both fields. A potential consequence of theta phase precession is that the cell with the smaller field would phase precess faster through the theta cycle than the other cell – initially, it would fire later in the theta cycle than the cell with a larger field, but as the animal moves towards the end of the small basis field it would fire earlier. These periods of potentiation and depression instigated by STDP could act against each other, and the extent to which they cancel each other out would depend on the relative placement of the two fields, their size difference, and the parameters of the learning rule. To test this, we simulated an

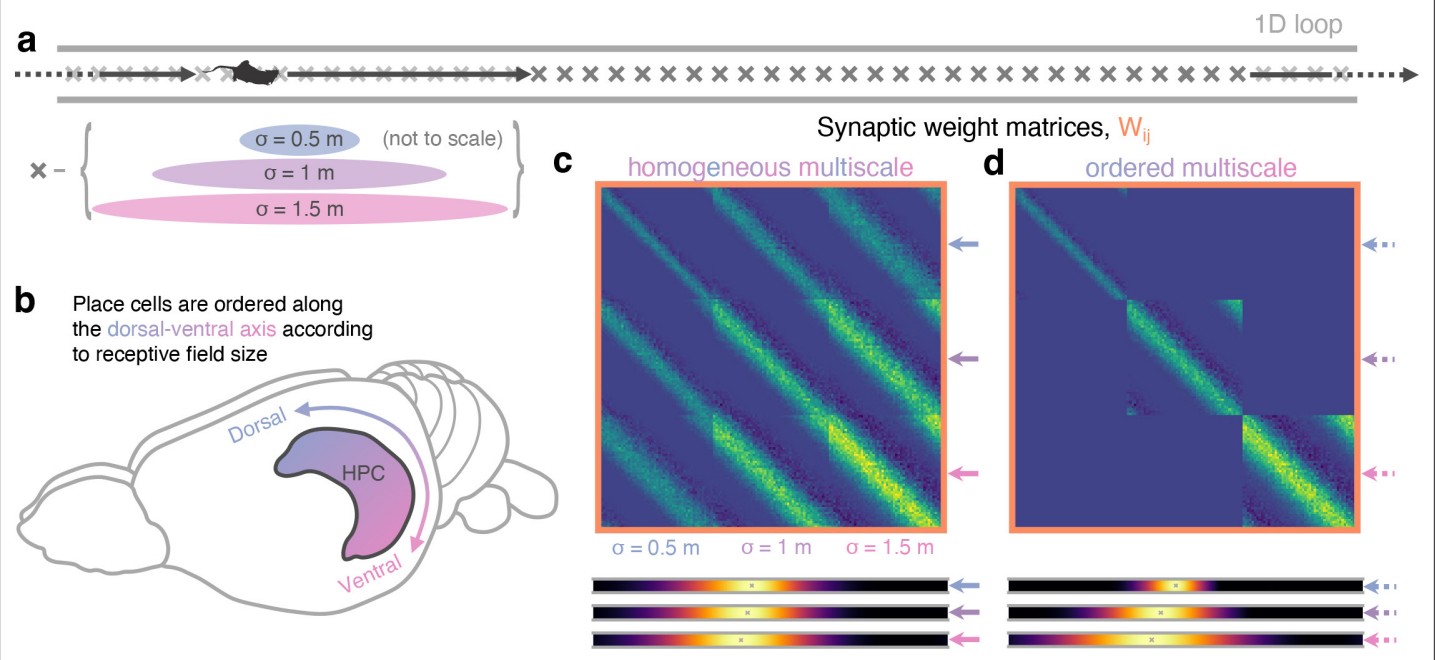

**Figure 4.** Multiscale successor representations are stored by place cells with multi-sized place fields but only when sizes are segregated along the dorso-ventral axis. (**a**) An agent explores a 1D loop maze with 150 places cells of different sizes (50 small, 50 medium, and 50 large) evenly distributed along the track. (**b**) In rodent hippocampus, place cells are observed to be ordered along the dorso-ventral axis according to their field size. (**c**) When cells with different field sizes are homogeneously distributed throughout hippocampus all postsynaptic successor features can bind to all presynpatic basis features, regardless of their size (top). Short timescale successor representations are overwritten, creating three equivalent sets of redundantly large-scale successor features (bottom). (**d**) Ordering cells leads to anatomical segregation; postsynaptic successor features can only bind to basis features in the same size range (off-diagonal block elements are zero) preventing cells with different size fields from binding. Now, three dissimilar sets of successor features emerge with different length scales, corresponding to successor features of different discount time horizons.

agent, learning according to our STDP model in the circular track environment, with, simultaneously, three sets of differently sized basis features ($\sigma = 0.5$, $1.0$ and $1.5$ m, *Figure 4a*). Such ordered variation in field size has been observed along the dorso-ventral axis of the hippocampus (*Kjelstrup et al., 2008*; *Strange et al., 2014*; *Figure 4b*), and has been theorised to facilitate successor representation predictions across multiple time-scales (*Stachenfeld et al., 2017*; *Momennejad and Howard, 2018*).

When we trained the STDP model on a population of homogeneously distributed multiscale basis features, the resulting weight matrix displayed binding across the different sizes regardless of the scale difference (*Figure 4c* top). This in turn leads to a population of downstream successor features with the same redundantly large scale (*Figure 4c* bottom). The negative interaction between different sized fields was not sufficient to prevent binding and, as such, the place fields of small features are dominated by contributions from bindings to larger basis features. Conversely, when these multiscale basis features were ordered along the dorso-ventral axis to prevent binding between the different scales – cells of the three scales were processed separately (*Figure 4d* top) – the multiscale structure is preserved in the resulting successor features (*Figure 4d* bottom). We thus propose that place cell size can act as a proxy for the predictive time horizon, $\tau$ – also called the discount parameter, $\gamma = e^{-\frac{dt}{\tau}}$, in discrete Markov Decision Processes. However, for this effect to be meaningful, plasticity between cells of different scales must be minimised to prevent short timescales from being overwritten by longer ones, this segregation may plausibly be achieved by the observed size ordering along the hippocampal dorsal-ventral axis.

## Discussion

Successor representations store long-run transition statistics and allow for rapid prediction of future states (*Dayan, 1993*) – they are hypothesised to play a central role in mammalian navigation strategies (*Stachenfeld et al., 2017*; *de Cothi and Barry, 2020*). We show that Hebbian learning between

spiking neurons, resembling the place fields found in CA3 and CA1, learns an accurate approximation to the successor representation when these neurons undergo phase precession with respect to the hippocampal theta rhythm. The approximation achieved by STDP explains a large proportion of the variance in the TD successor matrix and replicates hallmarks of successor representations (*Stachenfeld et al., 2014*; *Stachenfeld et al., 2017*; *de Cothi and Barry, 2020*) such as behaviourally biased place field skewing, elongation of place fields near walls, and clustering near doorways in both one and two-dimensional environments.

That the predictive skew of place fields can be accomplished with a STDP-type learning rule is a long-standing hypothesis; in fact, the authors that originally reported this effect also proposed a STDP-type mechanism for learning these fields (*Mehta et al., 2000*; *Mehta, 2001*). Similarly, the possible accelerating effect of theta phase precession on sequence learning has also been described in a number of previous works (*Jensen and Lisman, 1996*; *Koene et al., 2003*; *Reifenstein et al., 2021*). Until recently (*Fang et al., 2023*; *Bono et al., 2023*), SR models have largely not connected with this literature: they either remain agnostic to the learning rule or assume temporal difference learning (which has been well-mapped onto striatal mechanisms (*Schultz et al., 1997*; *Seymour et al., 2004*), but it is unclear how this is implemented in hippocampus) (*Stachenfeld et al., 2014*; *Stachenfeld et al., 2017*; *de Cothi and Barry, 2020*; *Geerts et al., 2020*; *Vértes and Sahani, 2019*). Thus, one contribution of this paper is to quantitatively and qualitatively compare theta-augmented STDP to temporal difference learning, and demonstrate where these functionally overlap. This explicit link permits some insights about the physiology, such as the observation that the biologically observed parameters for phase precession and STDP resemble those that are optimal for learning the SR (*Figure 2—figure supplement 3*), and that the topographic organisation of place cell sizes is useful for learning representations over multiple discount timescales (*Figure 4*). It also permits some insights for RL, such as that the approximate SR learned with theta-augmented STDP, while provably theoretically different from TD (Section: A theoretical connection between STDP and TD learning), is sufficient to capture key qualitative phenomena.

Theta phase precession has a dual effect not only *allowing* learning by compressing trajectories to within STDP timescales but also *accelerating* convergence to a stable representation by arranging the spikes from cells along the current trajectory to arrive in the order those cells are actually encountered (*Jensen and Lisman, 1996*; *Koene et al., 2003*). Without theta phase precession, STDP fails to learn a successor representation reflecting the current policy unless that policy is approximately unbiased. Further, by instantiating a population of place cells with multiple scales we show that topographical ordering of these place cells by size along the dorso-ventral hippocampal axis is a necessary feature to prevent small discount timescale successor representations from being overwritten by longer ones. Last, performing a grid search over STDP learning parameters, we show that those values selected by evolution are approximately optimal for learning successor representations. This finding is compatible with the idea that the necessity to rapidly learn predictive maps by STDP has been a primary factor driving the evolution of synaptic learning rules in hippocampus.

While the model is biologically plausible in several respects, there remain a number of aspects of the biology that we do not interface with, such as different cell types, interneurons and membrane dynamics. Further, we do not consider anything beyond the most simple model of phase precession, which directly results in theta sweeps in lieu of them developing and synchronising across place cells over time (*Feng et al., 2015*). Rather, our philosophy is to reconsider the most pressing issues with the standard model of predictive map learning in the context of hippocampus (e.g. the absence of dopaminergic error signals in CA1 and the inadequacy of synaptic plasticity timescales). We believe this minimalism is helpful, both for interpreting the results presented here and providing a foundation on which further work may examine these biological intricacies, such as whether the model's theta sweeps can alternately represent future routes (*Kay et al., 2020*) for example by the inclusion of attractor dynamics (*Chu et al., 2022*). Still, we show this simple model is robust to the observed variation in phase offsets between phase precessing CA3 and CA1 place cells across different stages of the theta cycle (*Mizuseki et al., 2012*). In particular, this phase offset is most pronounced as animals enter a field ($\sim 90°$) and is almost completely reduced by the time they leave it ($\sim 90°$; *Figure 2—figure supplement 4g*). Essentially, our model hypothesises that the majority of plasticity induced by STDP and theta phase precession will take place in the latter part of place fields, equating to earlier theta phases. Notably, this is in-keeping with experimental data showing enhanced coupling between CA3

and CA1 in these early theta phases (*Colgin et al., 2009*; *Hasselmo et al., 2002*). However, as our simulations show (*Figure 2—figure supplement 4g*), even if these assumptions do not hold true, the model is sufficiently robust to generate SR equivalent weight matrices for a range of possible phase offsets between CA3 and CA1.

Our model extends previous work – which required successor features to recursively expand in order to make long range predictions (e.g. as demonstrated in *Brea et al., 2016*; *Bono et al., 2023*) – by exploiting the existence of temporally compressed theta sweeps (*O'Keefe and Recce, 1993*; *Skaggs et al., 1996b*), allowing place cells with distant fields to bind directly without intermediaries or 'bootstrapping'. This configuration yields several advantages. First, learning with theta sweeps converges considerably faster than without them. Biologically, it is likely that successor feature learning via Hebbian learning alone (without theta precession) would be too slow to account for the rapid stabilisation of place cells in new environments at behavioural time scales (*Bittner et al., 2017*) – Dong et al. observed place fields in CA1 to increase in width for approximately the first 10 laps around a 3 m track (*Dong et al., 2021*). This timescale is well matched by our model with theta sweeps in which CA1 place cells reach 75% of their final extent after 5 min (or 9.6 laps) of exploration on a 5 m track but is markedly slower without theta sweeps.

Second, as well as extending previous work to large two-dimensional environments and complex movement policies our model also uses realistic population codes of overlapping Gaussian features. These naturally present a hard problem for models of spiking Hebbian learning since, in the absence of theta sweeps, the order in which features are encountered is not encoded reliably in the relative timing or order of their spikes at synaptic timescales. Theta sweeps address this by tending to sequence spikes according to the order in which their originating fields are encountered. Indeed our preliminary experiments show that when theta sweeps are absent the STDP successor features show little similarity to the TD successor features. Our work is thus particularly relevant in light of a recent trend to focus on biologically plausible features for reinforcement learning (*Gustafson and Daw, 2011*; *de Cothi and Barry, 2020*).

Other contemporary theoretical works have made progress on biological mechanisms for implementing the successor representation algorithm using somewhat different but complementary approaches. Of particular note are the works by *Fang et al., 2023*, who show a recurrent network with weights trained via a Hebbian-like learning rule converges to the successor representation in steady state, and *Bono et al., 2023* who derive a learning rule for a spiking feed-forward network which learns the SR of one-hot features by bootstrapping associations across time (see also *Brea et al., 2016*). Combined, the above models, as well as our own, suggest there may be multiple means of calculating successor features in biological circuits without requiring a direct implementation of temporal difference learning.

Our theory makes the prediction that theta contributes to *learning* predictive representations, but is not necessary to maintain them. Thus, inhibiting theta oscillations during exposure to a novel environment should impact the formation of successor features (e.g. asymmetric backwards skew of place fields) and subsequent memory-guided navigation. However, inhibiting theta in a familiar environment in which experience-dependent changes have already occurred should have little effect on the place fields: that is, some asymmetric backwards skew of place fields should be intact even with theta oscillations disrupted. To our knowledge, this has not been directly measured, but there are some experiments that provide hints. Experimental work has shown that power in the theta band increases upon exposure to novel environments (*Cavanagh et al., 2012*) – our work suggests this is because theta phase precession is critical for learning and updating stored predictive maps for spatial navigation. Furthermore, it has been shown that place cell firing can remain broadly intact in familiar environments even with theta oscillations disrupted by temporary inactivation or cooling (*Bolding et al., 2020*; *Petersen and Buzsáki, 2020*). It is worth noting, however, that even with intact place fields, these theta disruptions impair the ability of rodents to reach a hidden goal location that had already been learned, suggesting theta oscillations play a role in navigation behaviours even after initial learning (*Bolding et al., 2020*; *Petersen and Buzsáki, 2020*). Other work has also shown that muscimol inactivations to medial septum can disrupt acquisition and retrieval of the memory of a hidden goal location (*Chrobak et al., 1989*; *Rashidy-Pour et al., 1996*), although it is worth noting that these papers use muscimol lesions which Bolding and colleagues show also disrupt place-related firing, not just theta precession.

The SR model has a number of connections to other models from the computational hippocampal literature that bear on the interpretation of these results. A long-standing property of computational models in the hippocampus literature is a factorisation of spatial and reward representations (*Redish and Touretzky, 1998*; *Burgess et al., 1997*; *Koene et al., 2003*; *Hasselmo and Eichenbaum, 2005*; *Erdem and Hasselmo, 2012*), which permits spatial navigation to rapidly adapt to changing goal locations. Even in RL, the SR is also not unique in factorising spatial and reward representations, as purely model-based approaches do this too (*Dayan, 1993*; *Sutton and Barto, 1998*; *Daw, 2012*). The SR occupies a much more narrow niche, which is factorising reward from spatial representations while caching long-term occupancy predictions (*Dayan, 1993*; *Gershman, 2018*). Thus, it may be possible to retain some of the flexibility of model-based approaches while retaining the rapid computation of model-free learning.

A number of other models describe how physiological and anatomical properties of hippocampus may produce circuits capable of goal-directed spatial navigation (*Erdem and Hasselmo, 2012*; *Redish and Touretzky, 1998*; *Koene et al., 2003*). These models adopt an approach more characteristic of model-based RL, searching iteratively over possible directions or paths to a goal (*Erdem and Hasselmo, 2012*) or replaying sequences to build an optimal transition model from which sampled trajectories converge toward a goal (*Redish and Touretzky, 1998*) (this model bears some similarities to the SR that are explored by *Fang et al., 2023*, which shows dynamics converge to SR under a similar form of learning). These models rely on dynamics to compute the optimal trajectory, while the SR realises the statistics of these dynamics in the rate code and can therefore adapt very efficiently. Thus, the SR retains some efficiency benefits. These models are very well-grounded in known properties of hippocampal physiology, including theta precession and STDP, whereas until recently, SR models have enjoyed a much looser affiliation with exact biological mechanisms. Thus, a primary goal of this work is to explore how hippocampal physiological properties relate to SR learning as well.

More generally, in principle, any form of sufficiently ordered and compressed trajectory would allow STDP plasticity to approximate a successor representation. Hippocampal replay is a well documented phenomena where previously experienced trajectories are rapidly recapitulated during sharp-wave ripple events (*Wilson and McNaughton, 1994*), within which spikes show a form of phase precession relative to the ripple band oscillation (150–250 Hz; *Bush et al., 2022*). Thus, our model might explain the abundance of sharp-wave ripples during early exposure to novel environments (*Cheng and Frank, 2008*) – when new 'informative' trajectories, for example those which lead to reward, are experienced it is desirable to rapidly incorporate this information into the existing predictive map (*Mattar and Daw, 2018*).

The distribution of place cell receptive field size in hippocampus is not homogeneous. Instead, place field size grows smoothly along the longitudinal axis (from very small in dorsal regions to very large in ventral regions). Why this is the case is not clear – our model contributes by showing that, without this ordering, large and small place cells would all bind via STDP, essentially overwriting the short timescale successor representations learnt by small place cells with long timescale successor representations. Topographically organising place cells by size anatomically segregates place cells with fields of different sizes, preserving the multiscale successor representations. Further, our results exploring the effect of different phase offsets on STDP-successor learning (*Figure 2—figure supplement 4g*) suggest that the gradient of phase offsets observed along the dorso-ventral axis (*Lubenov and Siapas, 2009*; *Patel et al., 2012*) is insufficient to impair the plasticity induced by STDP and phase precession. The premise that such separation is needed to learn multiscale successor representations is compatible with other theoretical accounts for this ordering. Specifically, *Momennejad and Howard, 2018* showed that exploiting multiscale successor representations downstream, in order to recover information which is 'lost' in the process of compiling state transitions into a single successor representation, typically requires calculating the derivative of the successor representation with respect to the discount parameter. This derivative calculation is significantly easier if the cells – and therefore the successor representations – are ordered smoothly along the hippocampal axis.

Work in control theory has shown that the difficult reinforcement learning problem of finding an optimal policy and value function for a given environment becomes tractable if the policy is constrained to be near a 'default policy' (*Todorov, 2009*). When applied to spatial navigation, the optimal value function resembles the value function calculated using a successor representation for the default policy. This solution allows for rapid adaptation to changes in the reward structure since the successor

matrix is fixed to the default policy and need not be re-learnt even if the optimal policy changes. Building on this, recent work suggested the goal of hippocampus is not to learn the successor representation for the current policy but rather for a default diffusive policy (*Piray and Daw, 2021*).

Indeed, we found that in the absence of theta sweeps, the STDP rule learns a successor representation close to that of an unbiased policy, rather than the current policy. This is because without theta-sweeps to order spikes along the current trajectory, cells bind according to how overlapping their receptive fields are, that is, according to how close they are under a 'diffusive' policy. In this context it is interesting to note that a substantial proportion of CA3 place cells do not exhibit significant phase precession (*O'Keefe and Recce, 1993*; *Jeewajee et al., 2014*). One possibility is that these place cells with weak or absent phase precession might plausibly contribute to learning a policy-independent 'default representation', useful for rapid policy prediction when the reward structure of an environment is changed. Simultaneously, theta precessing place cells may learn a successor representation for the current (potentially biased) policy, in total giving the animal access to both an off-policy-but-near-optimal value function and an on-policy-but-suboptimal value function.

Finally, we comment on the approximate nature of the successor representations learnt by our biologically plausible model. The STDP successor features described here are unlikely to converge analytically to the TD successor features. Potentially, this implies that a value function calculated according to *Equation 31* would not be accurate and may prevent an agent from acting optimally. There are several possible resolutions to this point. First, the successor representation is unlikely to be a self contained reinforcement learning system. In reality, it likely interacts with other model-based or model-free systems acting in other brain regions such as nucleus accumbens in striatum (*Lisman and Grace, 2005*). Plausibly errors in the successor features are corrected for by counteracting adjustments in the reward weights implemented by some downstream model free error based learning system. Alternatively, it is likely that value function learnt by the brain is either fundamentally approximate or uses an different, less tractable, temporal discounting scheme. Ultimately, although in principle specialised and expensive learning rules might be developed to exactly replicate TD successor features in the brain, this maybe undesirable if a simple learning rule (STDP) is adequate in most circumstances. Indeed, animals – including humans – are known to act sub-optimally (*Zentall, 2015*; *de Cothi et al., 2022*), perhaps in part because of a reliance on STDP learning rules in order to learn long-range associations.

## Methods
### General summary of the model

The model comprises of an agent exploring a maze where its position $\mathbf{x}$ at time $t$ is encoded by the instantaneous firing of a population of $N$ CA3 basis features, $f_j(\mathbf{x}, t)$ for $j \in \{1, .., N\}$. Each has a spatial receptive field given by a thresholded Gaussian of peak firing rate 5 Hz:

$$f_j^x(\mathbf{x}(t)) = \begin{cases} \text{Gaussian}(\mathbf{x}_j, \sigma) - c & \text{if } \|\mathbf{x}(t) - \mathbf{x}_j\| < 1\text{m} \\ 0 & \text{otherwise} \end{cases} \tag{4}$$

where $\mathbf{x}_j$ is the location of the field peak, $\sigma = 1$m is the standard deviation and $c$ is a positive constant that keeps $f_j^x$ continuous at the threshold.

The theta phase of the hippocampal local field potential oscillates at 10 Hz and is denoted by $\phi_\theta(t) \in [0, 2\pi]$. Phase precession suppresses the firing rate of a basis features for all but a short period within each theta cycle. This period (and subsequently the time when spikes are produced, described in more details below) precesses earlier in each theta cycle as the agent crosses the spatial receptive field. Specifically, this is implemented by simply multiplying the spatial firing rate $f_j^x$ by a theta modulation factor which rises and falls according to a von Mises distribution in each theta cycle, peaking at a 'preferred phase', $\phi_j^*$, which depends on how far through the receptive field the agent has travelled (hence the spike timings implicitly encode location);

$$f_j^\theta(\phi_\theta(t)) = \text{VonMises}(\phi_i^*, \kappa) \tag{5}$$

where $\kappa = 1$ is the concentration parameter of the Von Mises distribution. These basis features in turn drive a population of $N$ downstream 'STDP successor features' (**Equation 2**).

Firing rates of both populations ($f_j(\mathbf{x}, \phi_\theta)$ and $\tilde{\psi}_i(\mathbf{x}, \phi_\theta)$) are converted to spike trains according to an inhomogeneous Poisson process. These spikes drive learning in the synaptic weight matrix, $W_{ij}$, according to an STDP learning rule (details below). In summary, if a presynaptic CA3 basis features fires immediately before a postsynaptic CA1 successor feature the binding strength between these cells is strengthened. Conversely if they fire in the opposite order, their binding strength is weakened.

For comparison, we also implement successor feature learning using a temporal difference (TD) learning rule, referred to as 'TD successor features', $\psi_i(\mathbf{x})$, to provide a ground truth against which we compare the STDP successor features. Like STDP successor features, these are constructed as a linear combination of basis features (**Equation 3**).

Temporal difference learning updates $M_{ij}$ as follows

$$M_{ij} \leftarrow M_{ij} + \eta \delta_{ij}^{\mathrm{TD}} \tag{6}$$

where $\delta_{ij}^{\mathrm{TD}}$ is the temporal difference error, which we derive below. In reinforcement learning the temporal difference error is used to learn discounted value functions (successor features can be considered a special type of value function). It works by comparing an unbiased sample of the true value function to the currently held estimate. The difference between these is known as the temporal difference error and is used to update the value estimate until, eventually, it converges on (or close to) the true value function.

## Definition of TD successor features and TD successor matrix
### Phase precession model details
In our hippocampal model CA3 place cells, referred to as basis features and indexed by $j$ and have thresholded Gaussian receptive fields. The threshold radius is $\sigma = 1$ m and peak firing rate is $F = 5$ Hz. Mathematically, this is written as

$$f_j^x(\mathbf{x}(t)) = \frac{F}{1 - e^{-\frac{1}{2}}}\left[e^{-\frac{\|\mathbf{x}(t) - \mathbf{x}_j\|^2}{2\sigma^2}} - e^{-\frac{1}{2}}\right]_+, \tag{7}$$

where $[f(x)]_+ = max(0, f(x))$, $\mathbf{x}_j$ is the centre of the receptive field and $\mathbf{x}(t)$ is the current location of the agent.

Phase precession is implemented by multiplying the spatial firing rate, $f_j^x(\mathbf{x})$, by a phase precession factor

$$f_j^\theta(\phi_\theta(t)) = 2\pi f_{\mathrm{VM}}\left(\phi_\theta(t)\Big|\phi_j^*(\mathbf{x}), \kappa\right). \tag{8}$$

where $f_{\mathrm{VM}}(x|\mu, \kappa)$ denotes the circular Von Mises distribution on $x \in (0, 2\pi]$ with mean $\mu = \phi_j^*(\mathbf{x})$ and spread parameter $\kappa = 1$. This factor is large only when the current theta phase,

$$\phi_\theta(t) = 2\pi\nu_\theta t \pmod{2\pi}, \tag{9}$$

which oscillates at $\nu_\theta = 10$ Hz, is close to the cell's 'preferred' theta phase,

$$\phi_j^*(\mathbf{x}(t)) = \pi + \beta\pi d_j(\mathbf{x}(t)). \tag{10}$$

$d_j(\mathbf{x}(t)) \in [-1, 1]$ tracks how far through the cell's spatial receptive field, as measured in units of $\sigma$, the agent has travelled:

$$d_j(\mathbf{x}(t)) = \frac{(\mathbf{x}(t) - \mathbf{x}_j) \cdot \frac{\dot{\mathbf{x}}(t)}{\|\dot{\mathbf{x}}(t)\|}}{\sigma}. \tag{11}$$

In instances where the agent travels directly across the centre of a cell (as is the case in 1D environments) then $(\mathbf{x}(t) - \mathbf{x}_j)$ and its normalised velocity (a vector of length 1, pointing in the direction of travel) $\frac{\dot{\mathbf{x}}(t)}{\|\dot{\mathbf{x}}(t)\|}$ are parallel such that $d_j(\mathbf{x})$ progresses smoothly in time from it's minimum, –1, to it's maximum, 1. In general, however, this extends to any arbitrary curved path an agent might take across the cell and matches the model used in **Jeewajee et al., 2014**. We fit $\beta$ and $\kappa$ to biological data in

Figure 5a of *Jeewajee et al., 2014* ($\beta = 0.5$, $\kappa = 1$). The factor of $2\pi$ normalises this term, although the instantaneous firing may briefly rise above the spatial firing rate $f_j^x(\mathbf{x})$, the average firing rate over the entire theta cycle is still given by the spatial factor $f_j^x(\mathbf{x})$. In total, the instantaneous firing rate of the basis feature is given by the product of the spatial and phase precession factors (*Equation 1*).

Note that the firing rate of a cell depends explicitly on its location through the spatial receptive field (its 'rate code') and implicitly on location through the phase precession factor (its 'spike-time code') where location dependence is hidden inside the calculation of the preferred theta phase. Notably, the effect of phase precession is only visible on rapid 'sub-theta' timescales. Its effect disappears when averaging over any timescale, $T_{av}$ substantially longer than theta timescale of $T_\theta = 0.1$ s:

$$\frac{1}{T_{av}} \int_t^{t+T_{av}} f_j(\mathbf{x}(t), \phi_\theta(t')) dt' \approx \frac{1}{T_{av}} \int_t^{t+T_{av}} f_j^x(\mathbf{x}(t')) dt' \qquad \text{for} \qquad T_{av} >> T_\theta \tag{12}$$

This is important since it implies that the effect of phase precession is only important for synaptic processes with very short integration timescales, for example, STDP.

Our phase precession model is 'independent' (essentially identical to *Chadwick et al., 2015*) in the sense that each place cell phase precesses independently from what the other place cells are doing. In this model, phase precession directly leads to theta sweeps as shown in *Figure 1*. Another class of models referred to as 'coordinated assembly' models (*Harris, 2005*) hypothesise that internal dynamics drive theta sweeps within each cycle because assemblies (aka place cells) dynamically excite one-another in a temporal chain. In these models, theta sweeps directly lead to phase precession. Feng and colleagues draw a distinction between theta precession and theta sequence, observing that while independent theta precession is evident right away in novel environments, longer and more stereotyped theta sequences develop over time (*Feng et al., 2015*). Since we are considering the effect of theta precession on the formation of place field shape, the independent model is appropriate for this setting. We believe that considering how our model might relate to the formation of theta sequences or what implications theta sequences have for this model is an exciting direction for future work.

## Synaptic learning via STDP

STDP is a discrete learning rule: if a presynaptic neuron $j$ fires before a postsynaptic neuron $i$ their binding strength $W_{ij}$ is potentiated, conversely if the postsynaptic neuron fires before the presynaptic then weight is depressed. This is implemented as follows.

First, we convert the firing rates to spike trains. We sample, for each neuron, from an inhomogeneous spike train with rate parameter $f_j(\mathbf{x}, t)$ (for presynaptic basis features) or $\tilde{\psi}_i(\mathbf{x}, t)$ for postsynaptic successor features. This is done over the period $[0, T]$ across which the animal is exploring.

$$\left( f_j(\mathbf{x}, t), [0, T] \right) \overset{Poisson}{\longmapsto} \{t_j^{\text{pre}}\} \qquad , \qquad \left( \tilde{\psi}_i(\mathbf{x}, t), [0, T] \right) \overset{Poisson}{\longmapsto} \{t_i^{\text{post}}\} \tag{13}$$

Asymmetric Hebbian STDP is implemented online using a trace learning rule. Each presynaptic spike from CA3 cell, indexed $j$, increments an otherwise decaying memory trace, $T_j^{\text{pre}}(t)$, and likewise an analogous trace for postsynaptic spikes from CA1, $T_i^{\text{post}}(t)$. We matched the STDP plasticity window decay times to experimental data: $\tau^{\text{pre}} = 20$ ms and $\tau^{\text{post}} = 40$ ms (*Bush et al., 2010*).

$$\tau^{\text{pre}} \frac{dT_j^{\text{pre}}(t)}{dt} = -T_j^{\text{pre}}(t) + \sum_{t' \sim \{t_j^{\text{pre}}\}} \delta(t - t') \tag{14}$$

$$\tau^{\text{post}} \frac{dT_i^{\text{post}}(t)}{dt} = -T_i^{\text{post}}(t) + \sum_{t' \sim \{t_i^{\text{post}}\}} \delta(t - t'). \tag{15}$$

We simplify our model by fixing weights during learning:

$$\tilde{\psi}_i(\mathbf{x}, t) = \sum_j W_{ij}^{\text{A}} f_j(\mathbf{x}, t) \qquad \text{During learning} \tag{16}$$

where we will refer to $W_{ij}^{\text{A}}$ as the "anchoring" weights which, up until now, have been set to the identity $W_{ij}^{\text{A}} = \delta_{ij}$. Since $f_j(\mathbf{x}, t)$ is the *phase precessing* features, $\tilde{\psi}_i(\mathbf{x}, t)$ also inherits phase precession from these features mapped through $W_{ij}^{\text{A}}$. Fixing the weights means that during learning the effect of changes in $W_{ij}$ are *not* propagated to the successor features (CA1), their influence is only considered

during post-learning recall broadly analogous to the distinct encoding and retrieval phases that have been hypothesised to underpin hippocampal function (*Hasselmo et al., 2002*). We relax this assumption in *Figure 2—figure supplement 2* and allow $W_{ij}$ to be updated online, showing this isn't essential.

After a period, $[0, T]$ of exploration the synaptic weights are updated on aggregate to account for STDP.

$$\text{W}_{ij}(T) = \text{W}_{ij}(0) + \eta \left[ a^{\text{pre}} \underbrace{\sum_{t_i \sim \{t_i^{\text{post}}\}} \delta(t - t_i) T_j^{\text{pre}}(t)}_{\text{"pre-before-post potentiations"}} + a^{\text{post}} \underbrace{\sum_{t_j \sim \{t_j^{\text{pre}}\}} \delta(t - t_j) T_i^{\text{post}}(t)}_{\text{"post-before-pre depressions"}} \right] \tag{17}$$

where the second terms accounts for the cumulative potentiation and depression due to STDP from spikes in the CA3 and CA1 populations. $\eta$ is the learning rate (here set to 0.01) and $a^{\text{pre}}$ and $a^{\text{post}}$ give the relative amounts of pre-before-post potentiation and post-before-pre depression, set to match experimental data from *Bi and Poo, 1998* as 1 and —0.4 respectively. The weights are initialised to the identity: $\text{W}_{ij}(0) = \delta_{ij}$.

Finally, when analysing the successor features after learning we use the updated weight matrix, not the anchoring weights, (and turn off phase precession since we are only interested in rate maps)

$$\tilde{\psi}_i(\mathbf{x}) = \sum_j \text{W}_{ij}(T) f_j^x(\mathbf{x}). \qquad \text{After learning} \tag{18}$$

## Temporal difference learning

To test our hypothesis that STDP is a good approximation to TD learning we simultaneously computed the *TD successor features* defined as the total expected future firing of a basis feature:

$$\psi_i(\mathbf{x}) = \mathbb{E}\left[ \int_t^\infty \frac{1}{\tau} e^{-\frac{t'-t}{\tau}} f_i^x(\mathbf{x}(t')) dt' \;\Big|\; \mathbf{x}(t) = \mathbf{x} \right]. \tag{19}$$

$\tau$ is the temporal discounting time-horizon (related to $\gamma$, the discount factor used in reinforcement learning on temporally discretised MDPs, $\gamma = e^{-\frac{dt}{\tau}}$) and the expectation is over trajectories initiated at position $\mathbf{x}$. This formula explains the one-to-one correspondence between CA3 cells and CA1 cells in our hippocampal model (*Figure 1b*): each CA1 cell, indexed $i$, learns to approximate the TD successor feature for its target basis feature, also indexed $i$. We set the discount timescale to $\tau = 4$ s to match relevant behavioural timescales for an animal exploring a small maze environment where behavioural decisions, such as whether to turn left or right, need to be made with respect to optimising future rewards occurring on the order of seconds.

We learn these successor features by tuning the weights of a linear decomposition over the basis feature set:

$$\psi_i(\mathbf{x}) = \sum_j \text{M}_{ij} f_j^x(\mathbf{x}), \tag{20}$$

this way we can directly compare $\text{M}_{ij}$ to the STDP weight matrix $\text{W}_{ij}$.

Our TD successor matrix, $\text{M}_{ij}$, should not be confused with the successor *representation* as defined in *Stachenfeld et al., 2017* and denoted $M(\mathsf{s}_i, \mathsf{s}_j)$, although they are analogous. $\text{M}_{ij}$ can be thought of as an analogue to $M(\mathsf{s}_i, \mathsf{s}_j)$ for spatially continuous (i.e. not one-hot) basis features, we show in the methods that they are equal (strictly, $M(\mathsf{s}, \mathsf{s}') = \text{M}_{ij}^\mathsf{T}$) in the limit of a discrete one-hot place cells.

## Temporal difference learning

The temporal difference (TD) update rule is used to learning the TD successor matrix (*Equation 20*). The standard TD(0) learning rule for a linear value function, $\psi_i(\mathbf{x})$, which basis feature weights $\text{M}_{ij}$ is (*Sutton and Barto, 1998*):

$$\text{M}_{ij} \leftarrow \text{M}_{ij} + \eta \delta_i f_j^x(\mathbf{x}) \tag{21}$$

where $\delta_i$ is the observed TD-error for the $i^{\text{th}}$ successor feature and $\eta$ is the learning rate. Note that we are only considering the spatial component of the firing rate, $f_j^x(\mathbf{x})$, not the phase modulation component, $f_j^\theta(\mathbf{x})$, which (as shown) would average away over any timescale significantly longer than the theta timescale (100ms). For now we will drop the superscript and write $f_j^x(\mathbf{x}) = f_j(\mathbf{x})$.

To find the TD-error, we must derive a temporally continuous analogue of the Bellman equation. Following *Doya, 2000*, we take the derivative of *Equation 19* which gives a consistency equation on the successor feature as follows:

$$\frac{d}{dt}\psi_i(\mathbf{x}(t)) = \frac{d}{dt}\int_t^\infty \frac{1}{\tau}e^{-\frac{t'-t}{\tau}}f_i(\mathbf{x}(t'))dt' \tag{22}$$

$$= \frac{1}{\tau}\left(\psi_i(\mathbf{x}(t)) - f_i(\mathbf{x}(t))\right) \tag{23}$$

This gives a continuous TD-error of the form

$$\delta_i(t) = \frac{d}{dt}\psi_i(\mathbf{x}(t)) + \frac{1}{\tau}\left(f_i(\mathbf{x}(t)) - \psi_i(\mathbf{x}(t))\right) \tag{24}$$

which can be rediscretised and rewritten by Taylor expanding the derivative ($\psi_i(t) = \frac{\psi_i(t) - \psi_i(t-dt)}{dt}$) to give

$$\delta_i(t) = \frac{1}{dt}\left(\frac{dt}{\tau}f_i(\mathbf{x}(t)) + \left(1 - \frac{dt}{\tau}\right)\psi_i(\mathbf{x}(t)) - \psi_i(\mathbf{x}(t-dt))\right). \tag{25}$$

This looks like a conventional TD-error term (typically something like $\delta_t = R_t + \gamma V_t - V_{t-1}$) except that we can choose $dt$ (the timestep between learning updates) freely. Finally expanding $\psi_i(\mathbf{x}(t))$ using (*Equation 3*) and substituting this back into *Equation 21* gives the update rule:

$$M_{ij} \leftarrow M_{ij} + \frac{\eta}{dt}\left[\frac{dt}{\tau}f_i(\mathbf{x}(t)) + \sum_k M_{ik}\left[\left(1 - \frac{dt}{\tau}\right)f_k(\mathbf{x}(t)) - f_k(\mathbf{x}(t-dt))\right]\right]f_j(\mathbf{x}(t)). \tag{26}$$

This rule does not stipulate a fixed time step between updates. Unlike traditional TD updates rules on discrete MDPs, $dt$ can take *any* positive value. The ability to adaptively vary $dt$ has potentially underexplored applications for efficient learning: when information density is high (e.g. when exploring new or complex environments, or during a compressed replay event *Skaggs and McNaughton, 1996a*) it may be desirable to learn regularly by setting $dt$ small. Conversely when the information density is low (for example in well known or simple environments) or learning is undesirable (for example the agent is aware that a change to the environment is transient and should not be committed to memory), $dt$ can be increased to slow learning and save energy. In practise, we set our agent to perform a learning update approximately every 1 cm along it's trajectory ($dt \approx 0.1$ s).

We add a small amount of L2 regularisation by adding the term $-2\eta\lambda M$ to the right hand side of *Equation 27*. This breaks the degeneracy in $M_{ij}$ caused by having a set of basis features which is overly rich to construct the successor features and can be interpreted, roughly, as a mild energy constraint favouring smaller synaptic connectomes. In total the full update rule from our TD successor matrix in matrix form is given by

$$M \leftarrow M + \frac{\eta}{dt}\left[\frac{dt}{\tau}\mathbf{f}(\mathbf{x}(t)) + M\left[\left(1 - \frac{dt}{\tau}\right)\mathbf{f}(\mathbf{x}(t)) - \mathbf{f}(\mathbf{x}(t-dt))\right]\right]\mathbf{f}^\mathsf{T}(\mathbf{x}(t)) - 2\eta\lambda M. \tag{27}$$

## Successor features in continuous time and space

Typically, as in *Stachenfeld et al., 2017*, the successor *representation* is calculated in discretised time and space. $M(\mathsf{s}_i, \mathsf{s}_j)$ encodes the expected discounted future occupancy of state $\mathsf{s}_j$ along a trajectory initiated in state $\mathsf{s}_i$:

$$M(\mathsf{s}_i, \mathsf{s}_j) = \mathbb{E}\left[\sum_{t=0}^\infty \gamma^t \delta(\mathsf{s}_t = \mathsf{s}_j) \;\middle|\; \mathsf{s}_0 = \mathsf{s}_i\right] \tag{28}$$

There are two forms of discretisation here. Firstly, time is discretised: it increases by a fixed increment,+1, to transition the state from $\mathsf{s}_t \rightarrow \mathsf{s}_{t+1}$. Secondly, assuming this is a spatial exploration task, space is discretised: the agent can be in exactly one state on any given time.

We loosen both these constraints reinstating time and space as continuous quantities. Since, for space, we cannot hope to enumerate an infinite number of locations, we represent the state by a population vector of diffuse, overlapping spatially localised place cells. Thus it is no longer meaningful

to ask what the expected future occupancy of a single location will be. The closest analogue, since the place cells are spatially localised, is to ask how much we expect place cell, $i$, centred at $\mathbf{x}_i$, to fire in the near (discounted) future. This continuous time constraint alters the sum over time into an integral over time. Further, the role of $\gamma$ which discounts state occupancy many time steps into the future, is replaced by $\tau$ which discounts firing a long time into the future. Thus the extension of the successor representation, $M(\mathsf{s}_i, \mathsf{s}_j)$, to continuous time and space is given by the successor *feature*,

$$\psi_i(\mathbf{x}) = \mathbb{E}\left[\int_t^\infty \frac{1}{\tau} e^{-\frac{t'-t}{\tau}} f_i\big(\mathbf{x}(t')\big) dt' \;\middle|\; \mathbf{x}(t) = \mathbf{x}\right]. \tag{29}$$

Why have we chosen to do this? Temporally it makes little sense to discretise time in a continuous exploration task: $\gamma$, the reinforcement learning discount factor, describes how many timesteps into the future the predictive encoding accounts for and so undesirably ties the predictive encoding to the otherwise arbitrary size of the simulation timestep, $dt$. In the continuous definition, $\tau$ intuitively describes how long into the future the predictive encoding discounts over and is independent of $dt$. This definition allows for online flexibility in the size of $dt$, as shown in *Equation 27*. This relieves the agent of a burden imposed by discretisation; namely that it must learn with a fixed time step,+1, all the time. Now the agent potentially has the ability to choose the fidelity over which to learn and this may come with significant benefits in terms of energy efficiency, as described above. Further, using the discretised form implicitly ties the definition of the successor representation (or any similarly defined value function) to the time step used in their simulation.

When space *is* discretised, the successor representation is a matrix encoding predictive relationships between these discrete locations. TD successor features, defined above, are the natural extension of the successor representation in a continuous space where location is encoded by a population of overlapping basis features, rather than exclusive one-hot states. The TD successor matrix, $M_{ij}$, can most easily be viewed as set of driving weights: $M_{ij}$ is large if basis feature $f_j(\mathbf{x})$ contributes strongly to successor feature $\psi_i(\mathbf{x})$. They are closely related (for example, in the effectively discrete case of non-overlapping basis features, it can be shown that the TD successor matrix then corresponds directly to the transpose of the successor representation, $M_{ij}^{\mathsf{T}} = M(\mathsf{s}_i, \mathsf{s}_i)$, see below for proof) but we believe the continuous case has more applications in terms of biological plausibility; electrophysiological studies show hippocampus encodes position using a population vector of overlapping place cells, rather than one-hot states. Furthermore the continuous case maps neatly onto known neural circuity, as in our case with CA3 place cells as basis features, CA1 place cells as successor features, and the successor matrix as the synaptic weights between them. In our case, the choice not to discretise space and use a more biologically compatible basis set of large overlapping place cells is necessary were our basis features to not overlap they would not be able to reliably form associations using STDP since often only one cell would ever fire in a given theta cycle.

For completeness (although this is not something studied in this report), this continuous successor feature form also allows for rapid estimation of the value function in a neurally plausible way. Whereas for the discrete case value can be calculated as:

$$V(\mathsf{s}_i) = \sum_j M(\mathsf{s}_i, \mathsf{s}_j) R(\mathsf{s}_j) \tag{30}$$

where $R(\mathsf{s}_j)$ is the per-time-step reward to be found at state $\mathsf{s}_j$, for continuous successor feature setting:

$$V(\mathbf{x}) = \sum_j \psi_j(\mathbf{x}) \mathsf{R}_j \tag{31}$$

where $\mathsf{R}_j$ is a vector of weights satisfying $\sum_j \mathsf{R}_j f_j(\mathbf{x}) = R(x)$ where $R(x)$ is the reward-rate found at location $\mathbf{x}$. (*Equation 31*) can be confirmed by substituting into it *Equation 29*. $\mathsf{R}_j$ (like $R(\mathsf{s}_j)$) must be learned independent to, and as well as, the successor features, a process which is not the focus of this study although correlates have been observed in the hippocampus (*Gauthier and Tank, 2018*). $V(\mathbf{x})$ is the temporally continuous value associated with trajectories initialised at $\mathbf{x}$:

$$V(\mathbf{x}) = \mathbb{E}\left[\int_t^\infty \frac{1}{\tau} e^{-\frac{t'-t}{\tau}} R\big(\mathbf{x}(t')\big) dt' \;\middle|\; \mathbf{x}(t) = \mathbf{x}\right]. \tag{32}$$

## Equivalence of the TD successor matrix to the successor representation

Here, we show the equivalence between $M(s_i, s_j)$ and $M_{ij}$. First we can rediscretise time by setting $dt'$ to be constant and defining $\gamma = 1 - \frac{dt'}{\tau}$ and $\mathbf{x}_n = \mathbf{x}(n \cdot dt')$. The integral in **Equation 29** becomes a sum,

$$\psi_i(\mathbf{x}) = (1 - \gamma)\mathbb{E}\left[\sum_{t=0}^{\infty} \gamma^t f_i(\mathbf{x}_t) \ \middle| \ \mathbf{x}_0 = \mathbf{x}\right]. \tag{33}$$

Next, we rediscretise space by supposing that CA3 place cells in our model have strictly non-overlapping receptive fields which tile the environment. For each place cell, $i$, there is continuous area, $\mathcal{A}_i$, such that for any location within this area place cell $i$ fires at a constant rate whilst all others are silent. When $\mathbf{x} \in \mathcal{A}_i$ we denote this state $s(\mathbf{x}) = s_i$ (since all locations in this area have identical population vectors).

$$f_i(\mathbf{x}) = \delta(\mathbf{x} \in \mathcal{A}_i) = \delta\big(s(\mathbf{x}) = s_i\big) \tag{34}$$

Let the initial state be $s(\mathbf{x}) = s_j$ (i.e. $\mathbf{x} \in \mathcal{A}_j$). Putting this into **Equation 33** and equating to **Equation 3**, the definition of our TD successor matrix, gives

$$\psi_i(\mathbf{x}) = \sum_k M_{ik}\delta(s_j = s_k) \quad = (1 - \gamma)\mathbb{E}\left[\sum_{t=0}^{\infty} \gamma^t \delta(s_t = s_i) \ \middle| \ s_0 = s_j\right], \tag{35}$$

confirming that

$$M_{ij}^{\mathsf{T}} \propto M(s_i, s_j). \tag{36}$$

## Simulation and analysis details

### Maze details

In the 1D open loop maze (**Figure 2a–e**), the policy was to always move around the maze in one direction (left to right, as shown) at a constant velocity of 16 cm s−1 along the centre of the track. Although figures display this maze as a long corridor, it is topologically identical to a loop; place cells close to the left or right sides have receptive fields extending into the right or left of the corridor respectively. Fifty Gaussian basis features of radius 1 m, as described above, are placed with their centres uniformly spread along the track. Agents explored for a total time of 30 min.

In the 1D corridor maze, **Figure 2f–j**, the situation is only changed in one way: the left and right hand edges of the maze are closed by walls. When the agent reaches the wall it turns around and starts walking the other way until it collides with the other wall. Agents explored for a total time of 30 min.

In the 2D two room maze, 200 basis feature are positioned in a grid across the two rooms (100 per room) then their location jittered slightly (**Figure 2k**). The cells are geodesic Gaussians. This means that the $\|\mathbf{x}(t) - \mathbf{x}_i\|^2$ term in **Equation 7** measures the distance from the agent location the centre of cell $i$ along the shortest walk which complies with the wall geometry. This explains the bleeding of the basis feature through the door in **Figure 3d**. Agents explored for a total time of 120 min.

The movement policy of the agent is a random walk with momentum. The agent moves forward with the speed at each discrete time step drawn from a Rayleigh distribution centred at 16 cm s−1. At each time step the agent rotates a small amount; the rotational speed is drawn from a normal distribution centred at zero with standard deviation 3 πrad s−1 ($\pi$ rad s−1 for the 1D mazes). Although the agent gets close to a wall (within 10 cm), the direction of motion is changed parallel to the wall, thus biasing towards trajectories which 'follow' the boundaries, as observed in real rats. This model was designed to match closely the behaviour of freely exploring rats and was adapted from the model initially presented in **Raudies and Hasselmo, 2012**. We add one additional behavioural bias: in the 2D two room maze, whenever the agent passes within 1 m of the centre point of the doorway connecting the two rooms, its rotational velocity is biased to turn it towards the door centre. This has the effect of encouraging room-to-room transitions, as is observed in freely moving rats (**Carpenter et al., 2015**).

## Analyses of the STDP and TD successor matrices

For the 1D mazes, there exists a translational symmetry relating the $N = 50$ uniformly distributed basis features and their corresponding rows in the STDP/TD weight matrices. This symmetry is exact for the 1D loop maze (all cells around a circle are rotated versions of one another) and approximate for the corridor maze (broken only for cells near to the left or right bounding wall). The result is that much the information in the linear track weight matrices *Figure 2b, c, g and h* can be viewed more easily by collapsing this matrix over the rows centred on the diagonal entry (plotted in *Figure 2d and i*). This is done using a circular permutation of each matrix row by a count, $n_i$, equal to how many times we must shift cell $i$ to the right in order for it's centre to lie at the middle of the track, $x_i = 2.5\text{m}$,

$$W_{ij}^{\text{aligned}} = W_{i,(j+n_i \ (\text{mod } 50))}. \tag{37}$$

This is the 'row aligned matrix'. Averaging over its rows removes little information thanks to the symmetry of the circular track. We therefore define the 1D quantity

$$\langle W \rangle_j := \frac{1}{N} \sum_{i=1}^{N} W_{ij}^{\text{aligned}}. \tag{38}$$

which is a convenient way to plot, in 1D, only the non-redundant information in the weight matrices.

## A theoretical connection between STDP and TD learning

Why does STDP between phase precessing place cells approximate TD learning? In this section, we attempt to shed some light on this question by analytically studying the equations of TD learning. Ultimately, comparisons between these learning rules are difficult since the former is inherently a discrete learning rule acting on pairs of spikes whereas the latter is a continuous learning rule acting on firing rates. Nonetheless, in the end we will draw the following conclusions:

1. In the first part, we will show that, under a small set of biologically feasible assumptions, temporal difference learning 'looks like' a spike-time dependent temporally asymmetric Hebbian learning rule (that is, roughly, STDP) where the temporal discount time horizon, $\tau$ is equal to the synaptic plasticity timescale $O(20\,\text{ms})$.
2. In the second part, we will see that this limitation that the temporal discount time horizon is restricted to the timescale of synaptic plasticity (i.e. very short) can be overcome by compressing the inputs. Phase precession, or more formally, theta sweeps, perform exactly the required compression.

In sum, there is a deep connection between TD learning and STDP and the role of phase precession is to compress the inputs such that a very short predictive time horizon amounts to a long predictive time horizon in decompressed time coordinates. We will finish by discussing where these learning rules diverge and the consequences of their differences on the learned representations. The goal here is not to derive a mathematically rigorous link between STDP and TD learning but to show that a connection exists between them and to point the reader to further resources if they wish to learn more.

## Reformulating TD learning to look like STDP

First, recall that the temporal difference (TD) rule for learning the successor features $\psi_i(\mathbf{x})$ defined in *Equation 19* takes the form:

$$\frac{d M_{ij}}{dt} = \eta \delta_i(t) e_j(t) \tag{39}$$

where $M_{ij}$ are the weights of the linear function approximator, *Equation 3* (Note, firstly, it is a coincidence specific to this study that the basis features of the linear function approximator, *Equation 3*, happen to be the same features of which we are computing the successor features, *Equation 19*. In general, this needn't be the case. Secondly, this analysis applies to *any* value function, not just successor features which are a specific example. If $f_i(\mathbf{x})$ in *Equation 19* was a reward density then $\psi_i(\mathbf{x})$ would become a true value function (discounted sum of future rewards) in the more conventional sense). and $\delta_i(t)$ is the continuous temporal difference error defined in *Equation 24*. $e_j(t)$ is the *eligibility trace* for feature $j$ defined according to

$$e_j(t) = \int_{-\infty}^{t} \frac{1}{\tau_e} e^{\frac{t-t'}{\tau_e}} f_j(\mathbf{x}(t')) dt' \tag{40}$$

or, equivalently, by its dynamics (which we will make use of)

$$e_j(t) = f_j(t) - \tau_e \dot{e}_j(t). \tag{41}$$

where $\tau_e \in [0, \tau]$ is a 'free' parameter, the eligibility trace timescale, analogous to $\lambda$ in discrete TD($\lambda$). When $\tau_e = 0$ we recover the learning rule we use to learn successor features, 'TD(0)', in **Equation 21**. Subbing **Equation 24** and **Equation 41** into this update rule, **Equation 39**, rearranges to give

$$\frac{dM_{ij}}{dt} = \eta \left( f_i e_j - \psi_i f_j + \tau \dot{\psi}_i e_j - \tau_e \psi_i \dot{e}_j \right) \tag{42}$$

where we redefined $\eta \leftarrow \eta' = \eta/\tau$. Now let the predictive time horizon be equal to the eligibility trace timescale. This setting is also called TD(1) or Monte Carlo learning,

$$\tau = \tau_e \tag{43}$$

Now

$$\frac{dM_{ij}}{dt} = \eta \left( f_i e_j - \psi_i f_j + \tau_e \frac{d}{dt}(\psi_i e_j) \right). \tag{44}$$

The final term in this update rule, the total derivative, can be ignored with respect to the stationary point of the learning process. To see why, consider the simple case of a periodic environment which repeats over a time period $T$ – this is true for the 1D experiments studied here. Learning is at a stationary point when the integrated changes in the weights vanish over one whole period:

$$0 = \int_t^{t+T} dt' \dot{M}_{ij}(t') = \eta \int_t^{t+T} dt' \left( f_i e_j - \psi_i f_j \right) + \eta \tau_e \int_t^{t+T} dt' \frac{d}{dt'}(\psi_i(t') e_j(t')) \tag{45}$$

$$= \eta \int_t^{t+T} dt' \left( f_i e_j - \psi_i f_j \right) + \eta \tau_e \left[ \psi_i(t+T) e_j(t+T) - \psi_i(t) e_j(t) \right] \tag{46}$$

$$= \eta \int_t^{t+T} dt' \left( f_i e_j - \psi_i f_j \right) \tag{47}$$

where the last term vanishes due to the periodicity. This shows that the learning rule converges to the same fixed point (i.e. the successor feature) irrespective of whether this term is present and it can therefore be removed. The dynamics of this updated learning rule won't strictly follow the same trajectory as TD learning but they will converge to the same point. Although strictly we only showed this to be true in the artificially simple setting of a periodic environment it is more generally true in a stochastic environment where the feature inputs depend on a stationary latent Markov chain (**Brea et al., 2016**).

Thus, a valid learning rule which converges onto the successor feature can be written as

$$\frac{dM_{ij}}{dt} = \eta \left( f_i(t) e_j(t) - \psi_i(t) f_j(t) \right) \tag{48}$$

Claim: this looks like a continuous analog of STDP acting on the weights between a set of input features, indexed $j$, and a set of downstream "successor features" indexed $i$. Each term in the above learning rule can be non-rigorously identified as follows, a key change is that the successor features neurons have two-compartments; a somatic compartment and a dendritic compartment:

- $f_i(t) := V_i^{soma}(t)$ is the somatic membrane voltage which is primarily set by a 'target signal'. In general, this target signal could be any reward density function, here it is the firing rate of the $i$th input feature.
- $\psi_i(t) := V_i^{dend}(t)$ is the voltage inside a dendritic compartment which is a weighted linear sum of the input currents, **Equation 3**. This compartment is responsible for learning the successor feature by adjusting its input weights, $M_{ij}$, according to **equation (48)**.
- $f_j(t) := I_j(t)$ are the synaptic currents into the dendritic compartment from the upstream features.
- $e_j(t) := \tilde{I}_j(t)$ are the low-pass filtered eligibility traces of the synaptic input currents.

$$\frac{dM_{ij}}{dt} = \eta \Big( \underbrace{V_i^{soma}(t) \tilde{I}_j(t)}_{\text{pre-before-post potentiation}} - \underbrace{V_i^{dend}(t) I_j(t)}_{\text{post-before-pre depression}} \Big) \tag{49}$$

This learning rule, mapped onto the synaptic inputs and voltages of a two-compartment neuron, is Hebbian. The first term potentiates the synapse $M_{ij}$ if there is a correlation between the low-pass filtered presynaptic current and the somatic voltage (which drives postsynaptic activity). More specifically this potentiation is is temporally asymmetric due to the second term which sets a threshold. A postsynaptic spike (e.g. when $V_i^{\text{soma}}(t)$ reaches threshold) will cause potentiation if

$$V_i^{\text{soma}}(t)\tilde{I}_j(t) > V_i^{\text{dend}}(t)I_j(t) \tag{50}$$

but since the eligibility trace decays uniformly after a presynaptic input this will only be true if the postsynaptic spike arrives very soon after. This is *pre-before-post* potentiation. Conversely an unpaired presynaptic input (e.g. when $I_j(t)$ spikes) will likely cause depression since this bolsters the second depressive term of the learning rule but not the first (note this is true if its synaptic weight is positive such that $V^{\text{dend}}(t)$ will be high too). This is analogous to *post-before-pre* depression. Whilst not identical, it is clear this rule bears the key hallmarks of the STDP learning rule used in this study, specifically: pre-before-post synaptic activity potentiates a synapse if post synaptic activity arrive within a short time of the presynaptic activity and, secondly, post-before-pre synaptic activity will typically result in depression of the synapse.

Intuitively, it now makes sense why asymmetric STDP learns successor features. If a postsynaptic spike from the ith neuron arrives just after a presynaptic spike from the jth feature it means, in all probability, that the presynaptic input features is 'predictive' of whatever caused the postsynaptic spike which in this case is the ith feature. Thus, if we want to learn a function which is predictive of the ith features future activity (its successor feature), we should increase the synaptic weight $M_{ij}$. Finally, identifying that this learning rule looks similar to STDP fixes the timescale of the eligibility trace to be the timescale of STDP plasticity i.e. $O(20 - 50\ \text{ms})$. And to derive this learning rule, we required that the temporal discount time horizon must equal the eligibility trace timescale, altogether:

$$\tau = \tau_{\text{e}} = \tau_{\text{STDP}} \approx 20 - 50\ \text{ms} \tag{51}$$

This limits the predictive time horizon of the learnt successor feature to a rather useless – but importantly non-zero – 20–50ms. In the next section, we will show how phase precession presents a novel solution to this problem.

## Theta phase precession compresses the temporal structure of input features

We showed in *Figure 1* how phase precession leads to theta sweeps. These phenomena are two sides of the same coin. Here we will start by positing the existence of theta sweeps and show that this leads to a potentially large amount of compression of the feature basis set in time.

First, consider two different definitions of position. $\mathbf{x}_T(t)$ is the 'True' position of the agent representing where it is in the environment at time $t$ is the 'Encoded' position of the agent which determines the firing rate of place cells which have spatial receptive fields $f_i(\mathbf{x}_E(t))$. During a theta sweep, the encoded position $\mathbf{x}_E(t)$ moves with respect to the true position $\mathbf{x}_T(t)$ at a relative speed of $\mathbf{v}_S(t)$ where the subscript $S$ distinguishes the 'Sweep' speed from the absolute speed of the agent $\dot{\mathbf{x}}_T(t) = \mathbf{v}_A(t)$. In total, accounting for the motion of the agent:

$$\dot{\mathbf{x}}_E(t) = \mathbf{v}_A(t) + \mathbf{v}_S(t) \tag{52}$$

Now consider how the population activity vector changes in time

$$\frac{d}{dt}f_i^T(\mathbf{x}_E(t)) = \nabla_\mathbf{x}f_i^T(\mathbf{x}) \cdot \dot{\mathbf{x}}_E(t) = \nabla_\mathbf{x}f_i^T(\mathbf{x}) \cdot (\mathbf{v}_A(t) + \mathbf{v}_S(t)) \tag{53}$$

and compare the time how it would varying in time if there was no theta sweep (i.e $\mathbf{x}_E(t) = \mathbf{x}_T(t)$)

$$\frac{df_i^T(\mathbf{x}_T(t))}{dt} = \nabla_\mathbf{x}f_i^T(\mathbf{x}) \cdot \frac{d\mathbf{x}_T(t)}{dt} = \nabla_\mathbf{x}f_i^T(\mathbf{x}) \cdot \mathbf{v}_A(t). \tag{54}$$

They are proportional. Specifically in 1D, where the sweep is observed to move in the same direction as the agent (from behind it to in front of it) this amount to compression of the temporal dynamics by a factor of

$$k_\theta = \frac{v_A + v_S}{v_A}. \tag{55}$$

This 'compression' is also true in 2D where sweeps are also observed to move largely in the same direction as the agent.

If this compression is large, it would solve the timescale problem described above. This is because learning a successor feature with a very small time horizon, $\tau$, where the input trajectory is heavily compressed in time by a factor of $\kappa_\theta$ amounts *to the same thing* as learning a successor feature with a long time horizon $\tau' = \tau\kappa_\theta$ where the inputs are not compressed in time.

What is $v_S$, and is it fast enough to provide enough compression to learn temporally extended SRs? We can make a very rough ballpark estimate. Data is hard to come by but studies suggest the intrinsic speed of theta sweeps can be quite fast. Figures in *Feng et al., 2015*, *Wang et al., 2020* and *Bush et al., 2022* show sweeps moving at up to, respectively, 9.4ms–1, 8.5ms–1 and 2.3ms–1. A conservative range estimate of $v_S \approx 5 \pm 5$ ms–1 accounts for very fast and very slow sweeps. The timescale of STDP is debated but a reasonable conservative estimate would be around $\tau_{\text{STDP}} \approx 35 \pm 15 \times 10^{-3}$ s which would cover the range of STDP timescales we use here. The typical speed of a rat, though highly variable, is somewhere in the range $v_A \approx 0.15 \pm 0.15$ ms–1. Combining these (with correct error analysis, assuming Gaussian uncertainties) gives an effective timescale increase of

$$\tau' = \tau k_\theta = \tau_{\text{STDP}} \frac{v_A + v_S}{v_A} \approx 1.1 \pm 1.7\text{s} \tag{56}$$

Therefore, we conclude theta sweeps can provide enough compression to lift the timescale of the SR being learn by STDP from short synaptic timescales to relevant behavioural timescales on the order of seconds. Note this ballpark estimate is not intended to be precise, and does not account for many unknowns for example the covariability of sweep speed with running speed[cite], variability of sweep speed with track length[cite] or cell size[cite] which could potentially extend this range further.

## Differences between STDP and TD learning: where our model does not work

We only drew a hand-waving connection between the TD-derived Hebbian learning rule in *Equation 48* and STDP. There are numerous difference between STDP and TD learning, these include the fact that:

1. Depression in *Equation 48* is dependent on the dendritic voltage which is not true for our STDP rule.
2. Depression in *Equation 48* is not explicitly dependent on the time between post and presynaptic activity, unlike STDP.
3. *Equation 48* is a continuous learning rule for continuous firing rates, STDP is a discrete learning rule applicable only to spike trains.

Analytic comparison is difficult due to this final difference which is why in this paper we instead opted for empirical comparison. Our goal was never to derive a spike-time dependent synaptic learning rule which replicates TD learning, other papers have done work in this direction (see *Brea et al., 2016*; *Bono et al., 2023*), rather we wanted to (i) see whether unmodified learning rules measured to be used by hippocampal neurons perform and (ii) study whether phase precession aids learning. Under regimes tested here, STDP seems to hold up well.

These differences aside, the learning rule does share other similarities to our model set-up. A special feature of this learning rule is that it postulates that somatic voltage driving postsynaptic activity during learning isn't affected by the neurons own dendritic voltage. Rather, dendritic voltages affect the *plasticity* by setting the potentiation threshold. These learning rules have been studies under the collective name of 'voltage dependent' Hebbian learning rules[CITE]. This matches the learning setting we use here where, during learning, CA1 neurons are driven by one and only one CA3 feature (the 'target feature') whilst the weights being trained $W_{ij}$ do not immediately effect somatic activity during learning. The lack of online updating matches the electrophysiological observation that plasticity between CA3 and CA1 is highest during the phase of theta when CA1 is driven by Entorhinal cortex and lowest at the phase when CA3 actually drives CA1 (*Hasselmo et al., 2002*).

Finally, there is one clear failure for our STDP model – learning very long timescale successor features. Unlike TD learning which can 'bootstrap' long timescale associations through intermediate connections, this is not possible with our STDP rule in its current form. *Brea et al., 2016* and *Bono et al., 2023* show how *Equation 48* can be modified to allow long timescale SRs whilst still enforcing the timescale constraint we imposed in *Equation 43* thus still maintaining the biological plausibility

of the learning rule, this requires allowing the dendritic voltage to modify the somatic voltage during learning in a manner highly similar to bootstrapping in RL. Specifically, in the former study, this is done by a direct extension to the two-compartment model, in the latter it is recast in a one-compartment model although the underlying mathematics shares many similarities. Ultimately both mechanisms could be at play; even in neurons endowed with the ability to bootstrap long timescale association with short timescale plasticity kernels phase precession would still increase learning speed significantly by reducing the *amount* of bootstrapping required by a factor of $\kappa_\theta$, something we intend to study more in future work. Finally it isn't clear what timescales predictive encoding in the hippocampus reach, there is likely to be an upper limit on the utility of such predictive representations beyond which the animal use model-based methods to find optimal solution which guide behaviour.

## Supplementary analysis

### *Figure 2—figure supplement 1*: Place cell size and movement statistics

For convenience, panel a of *Figure 2—figure supplement 1* duplicates the experiment shown in paper *Figure 2a–e*. The only change is learning time was extended from 30 minutes to 1 hour.

#### Movement speed variability

Panel b shows an experiment where we reran the simulation shown in paper *Figure 2a–e* except, instead of a constant motion speed, the agent moves with a variable speed drawn from a continuous stochastic process (an Ornstein Uhlenbeck process). The parameters of the process were selected so the mean velocity remained the same (16 cm s–1 left-to-right) but now with significant variability (standard deviation of 16 cm s–1 thresholded so the speed cannot go negative). Essentially, the velocity takes a constrained random walk. This detail is important: the velocity is not drawn randomly on each time step since these changes would rapidly average out with small $dt$, rather the change in the velocity (the acceleration) is random – this drives slow stochasticity in the velocity where there are extended periods of fast motion and extended periods of slow motion. After learning there is no substantial difference in the learned weight matrices. This is because both TD and STDP learning rules are able to average-over the stochasticity in the velocity and converge on representations representative of the mean statistics of the motion.

#### Smaller place cells and faster movement

Nothing fundamental prevents learning from working in the case of smaller place fields or faster movement speeds. We explore this in *Figure 2—figure supplement 1*, panel c, as follows: the agent speed is doubled from 16 cm s–1 to 32 cm s–1 and the place field size is shrunk by a factor of 5 from 2 m diameter to 40 cm diameter. To facilitate learning we also increase the cell density along the track from 10 cells m–1 to 50 cells m–1. We also shrink the track size from 5 m to 2 m (any additional track is redundant due to the circular symmetry of the set-up and small size of the place cells). We then train for 12 min. This time was chosen since 12 min moving at 32 cm s–1 on a 2 m track means the same number of laps as 60 min moving at 16 cm s–1 on a 5 m track (96 laps in total). Despite these changes the weight matrix converged with high similarity to the successor matrix with a shorter time horizon (0.5 s). Convergence time measured in minutes was faster than in the original case but this is mostly due to the shortened track length and increased speed. Measured in laps it now takes longer to converge due to the decreased number of spikes (smaller place fields and faster movement through the place fields). This can be seen in the shallower convergence curve, panel c (right) relative to panel a.

### *Figure 2—figure supplement 2*: Weight initialisation and updating schedule

#### Random initialisation

In *Figure 2—figure supplement 2*, panel a, we explore what happens if weights are initialised randomly. Rather than the identity, the weight matrix during learning is fixed ('anchored') to a sparse random matrix $W_{ij}^A$; this is defined such that each CA1 neuron receives positive connections from 3, 4, or 5 randomly chosen CA3 neurons with weights summing to one. In all other respects learning remains unchanged. CA1 neurons now have multi-modal receptive fields since they receive connections from multiple, potentially far apart, CA3 cells. This should not cause a problem since each

sub-field now acts as its own place field phase precessing according to whichever place cells in CA3 is driving it. Indeed it does not: after learning with this fixed but random CA3-CA1 drive, the synaptic weights are updated on aggregate and compares favourably to the successor matrix (panel a, middle and right). Specifically, this is the successor matrix which maps the unmixed uni-modal place cells in CA3 to the successor features of the new multi-modal 'mixed' features found in CA1 before learning. We note in passing that this is easy to calculate due to the linearity of the successor feature (SF): an SF of a linear sum of features is equal to a linear sum of SF, therefore we can calculate the new successor matrix using the same algorithm as before (described in the Methods) then rotating it by the sparse random matrix, $M'_{ij} = \sum_k W^A_{ik} M_{kj}$.

In order that some structure is visible matrix rows (which index the CA1 postsynaptic cells) have been ordered according to the location of the CA1 peak activity. This explains why the random sparse matrix (panel a, middle) looks ordered even though it is not. After learning the STDP successor feature looks close in form to the TD successor feature and both show a shift and skew backwards along the track (panel a, rights, one example CA1 field shown).

## Online weight updating

In *Figure 2—figure supplement 2*, panels b, c and d, we explore what happens if the weights are updated online during learning. It is not possible to build a stable fully online model (as we suspect the review realised) and it is easy to understand why: if the weight matrix doing the learning is also the matrix doing the driving of the downstream features then there is nothing to prevent instabilities where, for example, the downstream feature keeps shifting backwards (no convergence) or the weight matrix for some/all features disappears or blows up (incorrect convergence). However, it is possible to get most of the way there by splitting the driving weights into two components. The first and most significant component is the STDP weight matrix being learned online, this creates a 'closed loop' where changes to the weights affects the downstream features which in turn affect learning on the weights. The second smaller component is what we call the 'anchoring' weights, which we set to a fraction of the identity matrix (here $\frac{1}{2}$) and are not learned. In summary, *Equation 16* becomes

$$\tilde{\psi}_i(\mathbf{x}, t) = \sum_j \left( W_{ij}(t) + W^A_{ij} \right) f_j(\mathbf{x}, t) \tag{57}$$

for $W^A_{ij} = \frac{1}{2}\delta_{ij}$.

These anchoring weights provide structure, analogous to a target signal or 'scaffold' onto which the successor features will learn without risk of infinite backwards expansion or weight decay. After learning when analysing the weight/successor features the anchoring component is not considered.

Every other model of TD learning implicitly or explicitly has a form of anchoring. For example in classical TD learning each successor feature receives a fixed 'reward' signal from the feature it is learning to predict (this is the second term in *Equation 23* of our methods). Even other 'synaptically plausible' models include a non-learnable constant drive [see (*Bono et al., 2023*) CA3-CA1 model, more specifically the bias term in their Equation 12]. This is the approach we take here. We add the additional constraint that the sum of each row of the weight matrix must be smaller than or equal to 1, enforced by renormalisation on each time step. This constraint encodes the notion that there may be an energetic cost to large synaptic weight matrices and prevents infinite growth of the weight matrix.

$$W_{ij}(t) \leftarrow \frac{W_{ij}(t)}{\max(1, \sum_j W_{ij})} \tag{58}$$

The resulting evolution of the learnable weight component, $W_{ij}(t)$, is shown in panel b (middle shows row aligned averages of $W_{ij}(t)$ from t=0 minutes to to = 64 min, on the full matrices are shown) and panel f (full matrix) from being initialised to the identity. The weight matrix evolves to look like a successor matrix (long skew left of diagonal, negative right of diagonal). One risk, when weights are updated online, is that the asymmetric expansion continues indefinitely. This does not happen and the matrix stabilises after 15 min (panel e, colour progression). It is important to note that the anchoring component is smaller than the online weight component and we believe it could be made very small in the limit of less noisy learning (e.g. more cells or higher firing rates).

In panel c, we explore the combination: random weight initialisation *and* online weight updating. As can be seen, even with rather strong random initial weights learning eventually 'forgets' these and settles to the same successor matrix form as when identity initialisation was used.

In panel d, we show that anchoring *is* essential. Without it ($W^A_{ij} = 0$) the weight matrix initially shows some structure shifting and skewing to the left but this quickly disintegrates and no observable structure remains at the end of learning.

## Many-to-few spiking model

In *Figure 2—figure supplement 2*, panel e, we simulate the more biologically realistic scenario where each CA1 neuron integrates spikes (rather than rates) from a large (rather than equal) number of upstream CA3 neurons. This is done with two changes:

Firstly we increased the number of CA3 neurons from 50 to 500 while keeping the number of CA1 neurons fixed. Each CA1 neuron is now receives fixed anchoring drive from a Gaussian-weighted sum of the 10 (as opposed to 1) closest CA3 neurons.

Secondly, since in our standard model spikes are used for learning but neurons communicate via their rates, we change this so that CA3 spikes directly drive CA1 spikes in the form of a reduced spiking model. Let $X^{CA1}_{i,t}$ be the spike count of the $i^{th}$ CA1 neuron at timestep t and $X^{CA3}_{j,t}$ the equivalent for the $j^{th}$ CA3 neuron then, under the reduced spiking model,

$$\Pr(X^{CA1}_{i,t} = k) \quad = \text{Poisson}(k, \lambda_{i,t}) \tag{59}$$

$$\lambda_{i,t} \quad = \frac{1}{dt} \sum_j W^A_{ij} X^{CA3}_{j,t} \tag{60}$$

As can be expected, this model is very similar to the original model since CA3 spikes are noisy sample of their rates. This noise should average out over time and the simulations indeed confirm this.

## *Figure 2—figure supplement 3*: Hyperparameter sweep

We perform a hyperparameter sweep over STDP and phase precession parameters to see which are optimal for learning successor matrices. Remarkably the optimal parameters (those giving highest R2 between the weight matrix and the successor matrix) are found to be those – or vary close to those – used by biological neurons (*Figure 2—figure supplements 2 and 3*). Specifically, to avoid excess computational costs two independent sweeps were run: the first was run over the four relevant STDP parameters (the two synaptic plasticity timescales, the ratio of potentiation to depression and the firing rate) and the second was run over the phase precession parameters (phase precession spread parameter and the phase precession fraction).

On all cases, the optimal parameter sits close to the biological parameter we used in this paper (panel c, d). One exception is the firing rate where higher firing rates always giver better scores, likely due to the decreased effect of noise, however it is reasonable biology can't achieve arbitrarily high firing rates for energetic reasons.

## *Figure 2—figure supplement 4*: Phase precession

### The optimality of biological phase precession parameters

In *Figure 2—figure supplement 3*, we ran a hyperparameter sweep over the two parameters associated with phase precession: $\kappa$, the von Mises parameter describing how noisy phase precession is and $\beta$, the fraction of the full $2\pi$ theta cycle phase precession crosses. The results show that for both of these parameters there is a clear "goldilocks" zone around the biologically fitted parameters we chose originally. When there is too much (large $\kappa$, large $\beta$) or too little (small $\kappa$, small $\beta$) phase precession performance is worse than at intermediate biological amounts of phase precession. Whilst – according to the central hypothesis of the paper – it makes sense that weak or non-existence phase precession hinders learning, it is initially counter intuitive that strong phase precession also hinders learning.

We speculate the reason is as follows, when $\beta$ is too big phase precession spans the full range from 0 to $2\pi$, this means it is possible for a cell firing very late in its receptive field to fire just before a cell a long distance behind it on the track firing very early in the cycle because $2\pi$ comes just before 0 on the unit circle. When $\kappa$ is too big, phase precession is too clean and cells firing at opposite ends of the theta cycle will never be able to bind since their spikes will never fall within a 20ms window of

each other. We illustrate these ideas in *Figure 2—figure supplement 4* by first describing the phase precession model (panel a) then simulating spikes from 4 overlapping place cells (panel b) when phase precession is weak (panel c), intermediate/biological (panel d) and strong (panel e). We confirm these intuitions about why there exists a phase precession 'goldilocks' zone by showing the weight matrix compared to the successor matrix (right hand side of panels c, d and e). Only in the intermediate case is there good similarity.

## Phase precession of CA1

In most results shown in this paper, the weights are anchored to the identity during learning. This means each CA1 cells inherits phase precession from the one and only one CA3 cell it is driven by. It is important to establish whether CA1 still shows phase precession *after* learning when driven by multiple CA3 cells or, equivalently, during learning when the weights aren't anchored and it is therefore driven by multiple CA3 neurons. Analysing the spiking data from CA1 cells after learning (phase precession turned on) shows it does phase precession. This phase precession is noisier than the phase precession of a cell in CA3 but only slightly and compares favourably to real phase precession data for CA1 neurons (panel f, right, adapted from *Jeewajee et al., 2014*).

The reason for this is that CA1 cells are still localised and therefore driven mostly by cells in CA3 which are close and which peak in activity together at a similar phase each theta cycle. As the agent moves through the CA1 cell it also moves through all the CA3 cells and their peak firing phase precesses driving an earlier peak in the CA1 firing. Phase precession is CA1 after learning is noisier/broader than CA3 but far from non-existent and looks similar to real phase precession data from cells in CA1.

## Phase shift between CA3 and CA1

In *Figure 2—figure supplement 4g*, we simulate the effect of a decreasing phase shift between CA3 and CA1. As observed by *Mizuseki et al., 2012*, there is a phase shift between CA3 and CA1 neurons starting around 90 degrees at the end of each theta cycle (where cells fire as their receptive field is first entered) and decreasing to 0 at the start. We simulate this by adding a temporal delay to all downstream CA1 spikes equivalent to the phase shifts of 0°, 45°and 90°. The average of the weight matrices learned over all three examples still displays clear SR-like structure.

# Acknowledgements

We thank Wellcome for supporting this work through the Senior Research Fellowship awarded to C.B. [212281/Z/18/Z]. We also thank Samuel J Gershman and Talfan Evans for useful feedback on the manuscript.

# Additional information

### Competing interests

Kimberly L Stachenfeld: is affiliated with DeepMind. The author has no financial interests to declare. The other authors declare that no competing interests exist.

### Funding

| Funder | Grant reference number | Author |
| --- | --- | --- |
| Wellcome Trust | 212281/Z/18/Z | William de Cothi |

The funders had no role in study design, data collection and interpretation, or the decision to submit the work for publication. For the purpose of Open Access, the authors have applied a CC BY public copyright license to any Author Accepted Manuscript version arising from this submission.

### Author contributions

Tom M George, Conceptualization, Software, Formal analysis, Validation, Investigation, Visualization, Methodology, Writing – original draft, Writing – review and editing; William de Cothi, Conceptualization,

Formal analysis, Investigation, Methodology, Writing – original draft, Writing – review and editing; Kimberly L Stachenfeld, Supervision, Methodology, Writing – original draft, Writing – review and editing; Caswell Barry, Conceptualization, Supervision, Methodology, Writing – original draft, Project administration, Writing – review and editing

**Author ORCIDs**
Tom M George ⓘ http://orcid.org/0000-0002-4527-8810
William de Cothi ⓘ http://orcid.org/0000-0001-5624-9196
Kimberly L Stachenfeld ⓘ http://orcid.org/0000-0001-6936-4257
Caswell Barry ⓘ http://orcid.org/0000-0001-6718-0649

**Decision letter and Author response**
Decision letter https://doi.org/10.7554/eLife.80663.sa1
Author response https://doi.org/10.7554/eLife.80663.sa2

## Additional files

### Supplementary files
• MDAR checklist

### Data availability
All code associated with this project can be found at https://github.com/TomGeorge1234/STDP-SR, (*George, 2023*, copy archived at swh:1:rev:f126330b993d50cee021b1c356077bdab80299f4). There are no raw or external datasets associated with this project.

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
