## [Editor Report]

This theoretical work is important in that it bridges neural mechanisms within the hippocampus with the abstract computations it is thought to support for reinforcement learning. The study offers a potential mechanism by which spike timing dependent plasticity and theta phase precession within spiking neurons in CA3 and CA1 can yield successor representations. The simulations are compelling in that they continue to hold even when some of the simple but less realistic assumptions are relaxed in support of more realistic scenarios consistent with biological data.

---

## [Decision Letter]

**Decision letter after peer review:**

Thank you for submitting your article "Rapid learning of predictive maps with STDP and theta phase precession" for consideration by *eLife*. Your article has been reviewed by 2 peer reviewers, and the evaluation has been overseen by a Reviewing Editor and Michael Frank as the Senior Editor. The following individual involved in the review of your submission has agreed to reveal their identity: Michael E. Hasselmo (Reviewer #1).

Essential revisions:

1) Significantly more discussion of the work's relationship to relevant prior models of the hippocampus (as described by Reviewer #1)

2) New simulations that address Reviewer 2's concerns about biological plausibility.

3) Analysis that sheds light on why theta sequences + STDP approximates the TD algorithm (as described by Reviewer #2).

The second essential revision above may involve significant restructuring of the modeling approach. If the authors wish to undertake this, we will be happy to consider the substantially revised version for publication in *eLife*.

*Reviewer #1 (Recommendations for the authors):*

Page 4 – top line – "in the successor representation this is because CA3 place cells to the left…". I think this is confusing as the STDP model essentially generates the same effect. I think this should say: "In the network trained by Temporal Difference learning this is because CA3 place cells to the left…". This better description is used further down where the text says "between STDP and TD weight matrices". Throughout the manuscript

Page 4 – end of the first paragraph – "potentially becoming negative" – it is disconcerting to have this discussion of the idea of synaptic weights going from positive to negative in the context of the STDP model. One of the main advantages of this model is its biological realism, so it should not so casually mention violating Dale's law and having the synapse magically switch from being glutamatergic to GABAergic. This is disturbing to a neuroscientist.

Page 4- "is an essential element of this process." – The importance of theta phase precession to sequence learning with STDP has been discussed in numerous previous papers. For example, in a series of four papers in 1996, Jensen and Lisman describe in great detail a buffer mechanism for generating theta phase precession, and show how this allows encoding of a sequence. This is also explicitly discussed in Koene, Gorchetnikov, Cannon, and Hasselmo, Neural Networks, 2003, in terms of a spiking window of LTP less than 40 msec that requires a short-term memory buffer to allow spiking within this window.

Page 4 – "our model and the successor representation" – again this is confusing and should instead contrast "our model and the TD trained successor representation"

Page 6 – "in observed" – is observed.

Page 6 – "binding across the different sizes" – This needs to be stated more clearly in the text as it is very vague. I would suggest adding the phrase: "regardless of the scale difference".

Figure 4D – "create a physical barrier" – this is very ambiguous as it recalls a physical barrier in the environment as between two rooms – should instead say "created an anatomical segregation".

Page 8 – "hallmarks of successor representations" – there should be citations for what paper shows these hallmarks of the successor representation.

Page 8 – "arrive in the order" – Here is a location where citations to previous papers on the use of a phase precession buffer to correctly time spiking for STDP should be added (i.e. Jensen and Lisman, 1996; Koene et al. 2003).

Page 8 – "via Hebbian learning alone" – add "without theta phase precession" to be clear about what is not being included (since it could be anything such as other aspects of a learning rule).

Page 9 – "for spiking a feedforward network" – what does this mean – do they mean "for spiking in a feedforward network"? Aren't these other network mechanisms less biological realistic than the one presented here? I'd like to see some critical comparison between the models.

Page 9 – "makes a clear prediction…should impact subsequent navigation and the formation of successor features" – This is not a clear prediction but is instead circular – it essentially says – "if successor representations are not formed successor representations will not be observed" This is not much use to an experimentalist. This prediction should be stated in terms of a clear experimental prediction that refers only to physical testable quantities in an experiment and not circularly referring to the same vague and abstract concept of successor representations.

Page 9 – "to reach a hidden goal" – A completely different hippocampal modeling framework was used to model the finding of hidden goals in the Morris water maze in Erdem and Hasselmo, 2012, Eur. J. Neurosci and earlier work by Redish and Touretzky 1998, Neural Comp. To clarify the status of the successor representation framework relative to these older models that do not use successor representations, it would be very useful to have a few sentences of discussion about how the successor representation differs and is somehow either advantageous or biologically more realistic than these earlier models.

Page 9 "Lesions of the medial septum" – inactivation of the medial septum has also been shown to impair performance in Morris water maze (Chrobak et al. 2006).

Page 9 – "physical barrier to binding" – this is again very confusing as there is no physical barrier in the hippocampus. They should instead say "anatomical segregation"

Citation 32 – Mommenejad and Howard, 2018 – This is a very important citation and highly relevant to the discussion. However, I think it should just be cited as BioRXiv. It is confusing to call it a preprint.

*Reviewer #2 (Recommendations for the authors):*

This is an interesting study, and I enjoyed reading it. However, I have a number of concerns, particularly regarding the biological plausibility of the model, that I believe can be addressed with additional simulations and analysis.

– I had a number of concerns regarding the biological plausibility of the model and the choice of parameter settings, especially:

1) Mapping from rates to rates. The CA3 neurons act on CA1 neurons via their firing rate rather than their spikes, but the STDP rule acts on the spikes. What happens if the CA1 neurons are driven by the synaptically-filtered CA3 spikes rather than the underlying rates? How does the model perform, and how does the performance vary with the number of CA3 neurons (since more neurons may be required in order to average over the stochastic spikes)?

2) Weights are initialised as Wij=deltaij, meaning a 1-1 correspondence from CA3 to CA1 cells. This would have been ok, except that the weights are not updated during learning – they are held fixed during the entire learning phase and only updated on aggregate after learning. Thus, during the entire learning process each CA1 cell is driven by exactly 1 CA3 cell, and therefore simply inherits (or copies) the activity of that CA3 cell (according to equation 2). If either 1) a more realistic weight initialisation were used (e.g., random) or 2) weights were updated online during learning, it seems likely that the proposed mechanism would no longer work.

3) Lack of discussion of phase precession in CA1 cells. What are the theta firing patterns of CA1 (successor) cells in the model? Do they exhibit theta sequences and/or phase precession? We are never told this. The spike phase of the downstream CA1 cell is extremely important for STDP, as it determines whether synapses associated with past or future events are potentiated or suppressed (see Figure 8 of Chadwick et al. 2016, *eLife*). Based on my understanding, in the current setup CA1 place cells should produce phase precession during learning (before weights are updated), but only because each CA1 cell copies the activity of exactly one CA3 cell, which is unrealistic. Moreover, after the weights are updated, whether they produce phase precession is no longer clear. It is important to determine whether the proposed mechanism works in the more realistic scenario in which both CA3 and CA1 cells exhibit phase precession, but CA1 cells are driven by multiple CA3 cells.

4) Related to the preceding comment, there is a phase shift/delay between CA3 and CA1 (Mizuseki, Buzsaki et al., 2010). This doesn't seem to have been taken into account. Can the model be set up so that i) CA1 cells receive inputs from multiple CA3 cells ii) both CA3 and CA1 cells exhibit phase precession iii) there is the appropriate phase delay between CA3 and CA1?

5) Dependence of learning on the noisiness of phase precession. The hyperparameter sweep seems to omit some of the most important variables, such as the spread paramaeter (kappa) and the place field width and running speed (see next comment). Since the successor representation is shown to be learned well when kappa=1 but not when kappa=0 (i.e. when phase precession is removed), this leaves open the question of what happens when kappa is bigger than or small than 1. It would be nice to see kappa systematically varied and the consequences explored.

6) Wide place fields and slow speeds. Place fields in the model have a diameter of 2 metres. This is quite big – bigger than typical place field sizes in the dorsal hippocampus (which often have around 30 cm diameter, or 15 cm radius). Moreover, the chosen velocity of 16 cm/s is quite slow, and rats often run much faster in experiments (30 cm/s and higher). With the chosen parameters, it takes the rodent 12.5 s to traverse a place field, which is unrealistically long. My concern is that this setup leads to a large number of spikes per pass through a place field and that this unrealistic setting is needed for the proposed mechanism to learn effectively in a reasonable number of laps. What happens when place fields are smaller and running speeds faster, as is typically found in experiments? How many laps are required for convergence?

7) Running speed-dependence of phase precession and firing rate. The rat is assumed to run at a fixed speed – what happens when speed is allowed to vary? Running speed has profound effects on the firing of place cells, including i) a change in their rate of phase precession ii) a change in their firing rate (Huxter et al., 2003). More simulations are needed in which running speed varies lap-by-lap, and/or within laps.

8) Two-dimensional phase precession. There is debate over how 2D environments are encoded in the theta phase (Chadwick et al. 2015, 2016; Huxter et al., 2008; Climer et al., 2013; Jeewajee et al., 2013). This should be mentioned and discussed – how much do the results depend on the specific assumptions regarding phase precession in 2D? For example, Huxter et al. found that, when animals pass through the edge of a place field, the cell initially precesses but then processes back to its initial phase, but this isn't captured by the model used in the present study. Chadwick et al. (2016) proposed a model of two-dimensional phase precession based on the phase locking of an oscillator, which reproduces the findings of Huxter et al. and makes different predictions for phase precession in two dimensions than the Jeewajee model used by the authors. It would be nice to test alternative models for 2D phase precession and determine how well they perform in terms of generating successor-like representations.

9) Modelling the distribution of place field sizes along the dorsoventral axis. Two important phenomena were omitted that are likely important and could alter the conclusions. First, there is a phase gradient along the dorsoventral axis, which generates travelling theta waves (Patel, Buszaki et al., 2012; Lebunov and Siapas, 2009). How do the results change when including a 180 (or 360) phase gradient along the DV axis? The authors state that "A consequence of theta phase precession is that the cell with the smaller field will phase precess faster through the theta cycle than the other cell – initially it will fire later in the theta cycle than the cell with a larger field, but as the animal moves towards the end of the small basis field it will fire earlier" – this neglects to consider the phase gradient along the DV axis (see also Leibold and Monsalve-Mecado, 2017). Second, the authors chose three discrete place field sizes for their dorsoventral simulations. How would these simulations look if a continuum of sizes were used reflecting the gradient along the dorsoventral axis? Going further, CA1 cells likely receive input from CA3 cells with a distribution of place field sizes rather than a single place field size – how would the model behave in that case?

– There is no theoretical analysis of why theta sequences+STDP approximates the TD algorithm, or when the proposed mechanism might/might not work. The model is simple enough that some analysis should be possible. It would be nice to see this elaborated on – can a reduced model be obtained that captures the learning algorithm embodied by theta sequences+STDP, and does this reduced model reveal an explicit link to the TD algorithm? If not, then why does it work, and when might it generalise/not work?

– The comparison of successor features to neural data was qualitative rather than quantitative, and often quite vague. This makes it hard to know whether the predictions of the model are actually consistent with real neural data. It would be much preferred if a direct quantitative comparison of the learned successor features to real data could be performed, for example, the properties of place fields near to doorways.

– Statistical structure of theta sequences. The model used by the authors is identical to that of Chadwick et al. (2015) (except for the thresholding of the Gaussian field), and so implicitly assumes that theta sequences are generated by the independent phase precession of each place cell. However, the authors mention in the introduction that other studies argue for the coordination of place cells, such that theta sequences can represent alternative futures on consecutive theta cycles (Kay et al.). This begs the question: how important is the choice of an independent phase precession model for the results of this study? For example, if the authors were to simulate a T-maze, would a model which includes cycling of alternative futures learn the successor representation better or worse than the model based on independent coding? Given that there now is a large literature exploring the coordination of theta sequences and their encoded trajectories, it would be nice to see some discussion of how the proposed mechanism depends on/relates to this.

[Editors' note: further revisions were suggested prior to acceptance, as described below.]

Thank you for resubmitting your work entitled "Rapid learning of predictive maps with STDP and theta phase precession" for further consideration by *eLife*. Your revised article has been evaluated by Michael Frank (Senior Editor) and the Reviewers.

The manuscript has been improved but there are some remaining issues that need to be addressed, as outlined below:

1. Spiking model. We all agree with you that a full spiking model would be much too complex. However, since you already generate spikes using a Poisson process, it would be useful to see a simulation where the Poisson rate of CA1 cell is determined by the integration of the incoming CA3 spikes (perhaps with many incoming CA3 neurons). If this doesn't work, you should discuss why this is the case and what the implications are for the model.

2. CA3 => CA1 projections. CA1 cells still receive input from just one CA3 cell for each place field in the updated model (at least in the majority of simulations). This allows precise theta timing of the pre and post -synaptic neurons which appears to be critical for the plasticity rule to function. For example, the mathematics of Geisler et al. 2007 shows that, if the CA1 cell would receive input from a set of phase precessing CA3 cells with spatially offset place field and a Gaussian weight profile (the most common way to model CA3-CA1 connections), then the CA1 cell would actually fire at the LFP theta frequency and wouldn't phase precess, and as a consequence the STDP mechanism would no longer learn the successor representation. This suggests strong constraints on the conditions under which the model can function which are currently not being adequately discussed. This should be investigated and discussed, and the constraints required for the model to function should be plainly laid out.

3. A similar concern holds with the phase offset between CA3 and CA1 found by Mizuseki et al. The theta+STDP mechanism learns the successor representation because the CA1 cells inherit their responses from a phase-precessing upstream CA3 cell, so the existence of a phase lag is troubling, because it suggests that CA1 cells are not driven causally by CA1 cells in the way the model requires. You may be right that, if some external force were to artificially impose a fixed lag between the CA3 and CA1 cell, the proposed learning mechanism would still function but now with a spatial offset. However, the Reviewer was concerned that the very existence of the phase lag challenges the basic spirit of the model, since CA1 cells are not driven by CA3 cells in the way that is required to learn causal relationships. At the very least, this needs to be addressed and discussed directly and openly in the Discussion section, but it would be better if the authors could implement a solution to the problem to show that the model can work when an additional mechanism is introduced to produce the phase lag (for example, a combination of EC and CA3 inputs at different theta phases?)

4. DV phase precession. The Reviewer would still like to see you introduce DV phase lags, which could be done with a simple modification of the existing simulations. At minimum, it is critical to remove/modify the sentence "A consequence of theta phase precession is that the cell with the smaller field will phase precess faster through the theta cycle than the other cell – initially it will fire later in the theta cycle than the cell with a larger field, but as the animal moves towards the end of the small basis field it will fire earlier." As R2 noted in their original review, this is not the case when DV phase lags are taken into account, as was shown by Leibold and Monsalve-Mercado (2017). Ideally, it would be best to update simulations updated to account for the DV phase lags and the discussion updated to account for their functional implications

*Reviewer #1 (Recommendations for the authors):*

I am satisfied with the response of the authors to the reviewer's comments.

*Reviewer #2 (Recommendations for the authors):*

While the reviewers have undertaken a number important additional analyses which address some of the concerns raised in the review, several of the most pressing concerns regarding biological plausibility have not been addressed. In particular, each CA1 place field is still inherited by exactly 1 CA3 place field in the updated protocol, and cells still interact via their firing rates with spikes only being used for the weight updates. Moreover, the authors chose not to address concerns regarding quantitative comparisons between the model and data. Overall, while the authors correctly point out that their primary contribution should be viewed as illustrating a mechanism to learn successor representations via phase precession and STDP, this message is undermined if the proposed mechanism can't function when reasonable assumptions are made regarding the number of cells and their mode of interaction.

Detailed points below:

1) In the updated protocol where CA1 cells receive inputs from multiple CA1 cells, the model still copies CA3 place fields to CA1 place fields in a 1-1 manner. This is not biologically plausible, since receptive fields in the brain are formed by integration of thousands of synaptic inputs from cells with spatially offset but overlapping receptive fields. Moreover, neurons in the model still interact from rates to rates, with plasticity instead acting only on spikes. The authors could have addressed these two concerns jointly by having CA1 cells integrate input from a large number of spiking CA3 neurons with spatially overlapping place fields and plastic synapses, but since the authors chose not to do so, I can only assume that the model doesn't work when realistic assumptions are incorporated. Such an approach needn't involve simulating a full spiking network as the authors suggest – rather, a GLM/LNP style model can be used to model CA1 spikes in response to CA3 spiking input. Moreover, I do not see any reason why this should complicate the comparison to the TD successor representation as suggested by the authors, as the model would still have a continuous rate underlying the Poisson process that could be used to this end. If the proposed model can't be made to work with realistic numbers of CA3 neurons (with realistic firing rates and plastic synapses), then the proposed mechanism is not a plausible learning rule for the hippocampus, which undercuts the central message of the study.

2) The authors chose not perform a quantitative comparison of the model to experimental data (e.g., clustering of place fields around doorways etc.), leaving a central concern unaddressed. While I understand that theories of the hippocampal successor representation more generally have been compared to data, the lack of quantitative comparison of the particular model proposed in this study is still troubling to me.

3) Many other concerns were not addressed, such as:

– The phase shift between CA3 and CA1. While the authors may be correct that, if a phase shift were artificially imposed on the model, this would entail a spatial shift along the track, the model as it stands is premised on the notion that CA1 cells inherit their activity entirely from upstream CA3 cells, and the model predicts that the two regions are in phase with one another. If a phase shift were imposed by another mechanism (e.g. EC input), then CA1 cells would no longer inherit their responses from CA3, and the proposed mechanism for learning the successor representation would no longer function. Thus, it seems essential to the proposed model that CA3 and CA1 are in phase, in contrast to experimental data.

– The phase shift along the DV axis and its impact on phase relationships. In the revised manuscript, the authors still say "A consequence of theta phase precession is that the cell with the smaller field will phase precess faster through the theta cycle than the other cell – initially it will fire later in the theta cycle than the cell with a larger field, but as the animal moves towards the end of the small basis field it will fire earlier.", but as pointed out in the original review (and shown by Leibold et al.), this is not true when the DV phase shift is included. I see no reason why unrealistic assumptions should be made in the model regarding DV phase precession.

---

## [Author Response]

Essential revisions:1) Significantly more discussion of the work's relationship to relevant prior models of the hippocampus (as described by Reviewer #1)

We have added a large quantity of text addressing the work’s relationship to relevant prior models of the hippocampus. We have added substantially to the introduction and discussion, and also have made other additions throughout the results to provide better context.

2) New simulations that address Reviewer 2's concerns about biological plausibility.

We have performed several new simulations, producing new results that speak to the model’s robustness and biological plausibility, constituting 3 entirely new multipanel supplementary figures examining the effects on the model of place field size, running speed, phase precession parameters, weight initialisation, weight update regimes and downstream phase precession in CA1.

3) Analysis that sheds light on why theta sequences + STDP approximates the TD algorithm (as described by Reviewer #2).

A significant new theoretical section provides mathematical insight as to why a combination of STDP and theta phase precession can approximate the temporal difference learning algorithm.

Reviewer #1 (Recommendations for the authors):Page 4 – top line – "in the successor representation this is because CA3 place cells to the left…". I think this is confusing as the STDP model essentially generates the same effect. I think this should say: "In the network trained by Temporal Difference learning this is because CA3 place cells to the left…". This better description is used further down where the text says "between STDP and TD weight matrices". Throughout the manuscript

Thank you for this suggestion. We’ve gone through the text and implemented this change where the issue arises, as well as adding the sentence clarifying our terms (described in the in response to the public review in response to point 4).

Page 4 – end of the first paragraph – "potentially becoming negative" – it is disconcerting to have this discussion of the idea of synaptic weights going from positive to negative in the context of the STDP model. One of the main advantages of this model is its biological realism, so it should not so casually mention violating Dale's law and having the synapse magically switch from being glutamatergic to GABAergic. This is disturbing to a neuroscientist.

Thank you for this valid point – we’ve added the following line to follow that sentence:

“So, for example, if a postsynaptic neuron reliably precedes its presynaptic cell on the track, the corresponding weight will be reduced, potentially becoming negative. We note that weights changing their sign is not biologically plausible, as it is a violation of Dale’s Law [43]. This could perhaps be corrected with the addition of global excitation or by recruiting inhibitory interneurons.”

Page 4- "is an essential element of this process." – The importance of theta phase precession to sequence learning with STDP has been discussed in numerous previous papers. For example, in a series of four papers in 1996, Jensen and Lisman describe in great detail a buffer mechanism for generating theta phase precession, and show how this allows encoding of a sequence. This is also explicitly discussed in Koene, Gorchetnikov, Cannon, and Hasselmo, Neural Networks, 2003, in terms of a spiking window of LTP less than 40 msec that requires a short-term memory buffer to allow spiking within this window.

We agree that the paper would benefit from better connection with the prior work on sequence learning with STDP and have added text to the introduction and discussion. In the introduction, we have added:

“One of the consequences of phase precession is that correlates of behaviour, such as position in space, are compressed onto the timescale of a single theta cycle and thus coincide with the time-window of STDP O(20 − 50 ms) [8, 18, 20, 21]. This combination of theta sweeps and STDP has been applied to model a wide range of sequence learning [22, 23, 24], and as such, potentially provides an efficient mechanism to learn from an animal’s experience – forming associations between cells which are separated by behavioural timescales much larger than that of STDP.”

And we’ve included a paragraph to the discussion to make this clear. This is contained in the paragraph above, in our response to point 1 in the public review (see paragraph starting “That the predictive skew of place fields can be accomplished…”).

Page 4 – "our model and the successor representation" – again this is confusing and should instead contrast "our model and the TD trained successor representation"

Thank you, we have made this change to the text.

Page 6 – "in observed" – is observed.

Thank you – fixed.

Page 6 – "binding across the different sizes" – This needs to be stated more clearly in the text as it is very vague. I would suggest adding the phrase: "regardless of the scale difference".

Thank you for the suggestion – we have implemented this change.

Figure 4D – "create a physical barrier" – this is very ambiguous as it recalls a physical barrier in the environment as between two rooms – should instead say "created an anatomical segregation".

Thank you for the suggestion – we have implemented this change.

Page 8 – "hallmarks of successor representations" – there should be citations for what paper shows these hallmarks of the successor representation.

Thank you – we have added citations to Stachenfeld et al. 2014, Stachenfeld et al. 2017, and de Cothi and Barry 2020 to this sentence.

Page 8 – "arrive in the order" – Here is a location where citations to previous papers on the use of a phase precession buffer to correctly time spiking for STDP should be added (i.e. Jensen and Lisman, 1996; Koene et al. 2003).

Thank you for the suggestion – we have implemented this change.

Page 8 – "via Hebbian learning alone" – add "without theta phase precession" to be clear about what is not being included (since it could be anything such as other aspects of a learning rule).

Thank you for the suggestion – we have implemented this change.

Page 9 – "for spiking a feedforward network" – what does this mean – do they mean "for spiking in a feedforward network"? Aren't these other network mechanisms less biological realistic than the one presented here? I'd like to see some critical comparison between the models.

Thank you for spotting this, this was actually a typo: the sentence should read “for a spiking feedforward network”, which in this case semantically alters the meaning.

Page 9 – "makes a clear prediction…should impact subsequent navigation and the formation of successor features" – This is not a clear prediction but is instead circular – it essentially says – "if successor representations are not formed successor representations will not be observed" This is not much use to an experimentalist. This prediction should be stated in terms of a clear experimental prediction that refers only to physical testable quantities in an experiment and not circularly referring to the same vague and abstract concept of successor representations.

We have addressed both of these points with changes to the same paragraph, so we have condensed them for readability. Firstly, we agree our stated “clear prediction” of the model was, in fact, unclear. We have rewritten the paragraph (see below) to clarify what we meant by this. Further, we were unable to locate the *Chrobak et al., 2006* reference, but found a *Chrobak et al., 1989* that matches this description. This is indeed relevant and we have added a citation (let us know if this was not the intended reference or if there is an additional relevant one):

Chrobak, J. J., Stackman, R. W., and Walsh, T. J. (1989). Intraseptal administration of muscimol produces dose-dependent memory impairments in the rat. *Behavioral and Neural Biology, 52*(3), 357–369. https://doi.org/10.1016/S0163-1047(89)90472-X

However, we noted that this paper uses a Muscimol inactivation to medial septum, which was shown by Bolding et al. 2019 to disrupt place-related firing as well as theta-band activity, so it is possible that the disruption to place code is what is driving the navigational deficit. Also, we accidentally referred to the inactivations performed by Bolding and colleagues as lesions, but in fact they performed temporary inactivations with a variety of drugs (tetracaine, muscimol, gabazine; the latter of which disrupted theta but left place-related firing intact).

We have modified our paragraph describing these points and the predictions of our model as follows:

“Our theory makes the prediction that theta contributes to learning predictive representations, but is not necessary to maintain them. Thus, inhibiting theta oscillations during exposure to a novel environment should impact the formation of successor features (e.g., asymmetric backwards skew of place fields) and subsequent memory-guided navigation. However, inhibiting theta in a familiar environment in which experience-dependent changes have already occurred should have little effect on the place fields: that is, some asymmetric backwards skew of place fields should be intact even with theta oscillations disrupted. To our knowledge this has not been directly measured, but there are some experiments that provide hints. Experimental work has shown that power in the theta band increases upon exposure to novel environments [62] – our work suggests this is because theta phase precession is critical for learning and updating predictive maps for spatial navigation. Furthermore, it has been shown that place cell firing can remain broadly intact in familiar environments even with theta oscillations disrupted by temporary inactivation or cooling [63, 64]. It is worth noting, however, that even with intact place fields, these theta disruptions impair the ability of rodents to reach a hidden goal location that had already been learned, suggesting theta oscillations play a role in navigation behaviours even after initial learning [63, 64]. Other work has also shown that muscimol inactivations to medial septum can disrupt acquisition and retrieval of the memory of a hidden goal location [65, 66], although it is worth noting that these papers use muscimol lesions which Bolding and colleagues show also disrupt place-related firing, not just theta precession.”

Page 9 – "to reach a hidden goal" – A completely different hippocampal modeling framework was used to model the finding of hidden goals in the Morris water maze in Erdem and Hasselmo, 2012, Eur. J. Neurosci and earlier work by Redish and Touretzky 1998, Neural Comp. To clarify the status of the successor representation framework relative to these older models that do not use successor representations, it would be very useful to have a few sentences of discussion about how the successor representation differs and is somehow either advantageous or biologically more realistic than these earlier models.

We agree this would be helpful, and have added the following text to the discussion:

“A number of other models describe how physiological and anatomical properties of hippocampus may produce circuits capable of goal-directed spatial navigation [30, 27, 23]. These models adopt an approach more characteristic of model- based RL, searching iteratively over possible directions or paths to a goal [30] or replaying sequences to build an optimal transition model from which sampled trajectories converge toward a goal [27] (this model bears some similarities to the SR that are explored by [40], which shows that under certain assumptions, dynamics converge to SR under a similar form of learning). These models rely on dynamics to compute the optimal trajectory, while the SR realises the statistics of these dynamics in the rate code and can therefore adapt very efficiently. Thus, the SR retains some efficiency benefits. The models cited above are very well-grounded in known properties of hippocampal physiology, including theta precession and STDP, whereas until recently, SR models have enjoyed a much looser affiliation with exact biological mechanisms. Thus, a primary goal of this work is to explore how hippocampal physiological properties relate to SR learning as well.”

Page 9 – "physical barrier to binding" – this is again very confusing as there is no physical barrier in the hippocampus. They should instead say "anatomical segregation".

Thank you for the suggestion – we have implemented this change as well.

Citation 32 – Mommenejad and Howard, 2018 – This is a very important citation and highly relevant to the discussion. However, I think it should just be cited as BioRXiv. It is confusing to call it a preprint.

Thank you for highlighting this, we have now changed the citation of this and all other cited preprints to their appropriate server e.g. bioRxiv.

Reviewer #2 (Recommendations for the authors):This is an interesting study, and I enjoyed reading it. However, I have a number of concerns, particularly regarding the biological plausibility of the model, that I believe can be addressed with additional simulations and analysis.

Thank you again for your thorough appraisal of our work. Your suggestions have led to new simulations and analyses that have contributed to a significantly improved manuscript. To briefly summarise, these include: 3 new multipanel supplementary figures examining the effects of place field size, running speed, phase precession parameters, weight initialisation,weight update regimes and CA1 phase precession; a new appendix providing theoretical analyses and insight into how and why the model approximates temporal difference learning; and an extension of the hyperparameter sweep analysis to include the parameters controlling phase precession.

– I had a number of concerns regarding the biological plausibility of the model and the choice of parameter settings, especially:1) Mapping from rates to rates. The CA3 neurons act on CA1 neurons via their firing rate rather than their spikes, but the STDP rule acts on the spikes. What happens if the CA1 neurons are driven by the synaptically-filtered CA3 spikes rather than the underlying rates? How does the model perform, and how does the performance vary with the number of CA3 neurons (since more neurons may be required in order to average over the stochastic spikes)?

We agree that swapping rates for spikes would move the model in the direction of being more biologically plausible; however, this ends up complicating the central comparison of the work. The purpose of this study was to test the hypothesis that a combination of STDP and theta phase precession can approximate the learning of successor representations via temporal difference (TD) learning. As such, since this TD learning rule applies to continuous firing rate values (e.g. de Cothi and Barry 2020), we find this mapping of rates to rates is an essential component to facilitate fair comparison between the two learning rules. This also simplifies our model and its interpretation, as it allows us to avoid the complexity of spiking models. However, we recognise that this is a biologically implausible assumption that we are making. An avenue for correcting this in future work would be to adopt the approach of Brea et al. 2016 or Bono et al. 2021 (on bioRxiv, also currently in review at *eLife*). We have now added the following text to the beginning of the Results section to clarify why this particular set up was used and its caveats:

“Further, the TD successor matrix Mij can also be used to generate the ‘TD successor features’ … allowing for direct comparison and analyses with the STDP successor features (Eqn. 2), using the same underlying firing rates driving the TD learning to sample spikes for the STDP learning. This abstraction of biological detail avoids the challenges and complexities of implementing a fully spiking network, although an avenue for correcting this would be the approach of Brea et al., 2016 and Bono et al., 2021 [41, 43].”

2) Weights are initialised as Wij=deltaij, meaning a 1-1 correspondence from CA3 to CA1 cells. This would have been ok, except that the weights are not updated during learning – they are held fixed during the entire learning phase and only updated on aggregate after learning. Thus, during the entire learning process each CA1 cell is driven by exactly 1 CA3 cell, and therefore simply inherits (or copies) the activity of that CA3 cell (according to equation 2). If either 1) a more realistic weight initialisation were used (e.g., random) or 2) weights were updated online during learning, it seems likely that the proposed mechanism would no longer work.

Thank you for this suggestion. Originally the 1-1 correspondence from CA3 to CA1 cells was to directly correspond to the definition of a successor feature (in which each successor feature corresponds to the predicted activity of a specific basis feature, e.g. Stachenfeld *et al.*, 2017; de Cothi and Barry 2020). However we acknowledge the biological implausibility of this approach. As such, we have updated the manuscript to include analyses of simulations where both the target CA1 activity is initialised by random weights (i.e. not the identity matrix), as well as where this target activity is updated online during learning (Figure 2—figure supplement 2). As we show, neither manipulation inhibits successful learning of the STDP successor features, with the caveat that when updating the target weights online, the target features need to be partially anchored to the external world to prevent perpetual drift in the target population. We now summarise these new simulations in the Results section:

“This effect is robust to variations in running speed (Figure 2—figure supplement 1 ) and field sizes (Figure 2—figure supplement 1), as well as scenarios where target CA1 cells have multiple firing fields (Figure 2–Supplement 2a) that are updated online during learning (Figure 2—figure supplement 2 ; see Supplementary Materials for more details)”

and elaborate on this method in the appendices/methods:

“Random initialisation: In Figure 2—figure supplement 2, panel a, we explore what happens if weights are initialised randomly. Rather than the identity, the weight matrix during learning is fixed (“anchored”) to a sparse random matrix WA ; this is defined such that each CA1 neuron receives positive connections from 3, 4 or 5 randomly chosen CA3 neurons with weights summing to one. […] After learning the STDP successor feature looks close in form to the TD successor feature and both show a shift and skew backwards along the track (panel a, rights, one example CA1 field shown).”

"Online weight updating: In Fig. 2 supplement 2, panels b, c and d, we explore what happens if the weights are updated online during learning. […] In panel d we show that anchoring is essential. Without it (WAij = 0) the weight matrix initially shows some structure shifting and skewing to the left but this quickly disintegrates and no observable structure remains at the end of learning.”

One interpretation of our set-up (the original one, described in the main text of the paper where weights are not updated online) is that it matches the “Separate Phases of Encoding and Retrieval Model” model [Hasselmo (2002)]. This paper describes how LTP between CA1 and CA3 synapses is strongest at the phase of theta when input to CA1 is primarily coming from entorhinal cortex. To quote the abstract of this paper: “effective encoding of new associations occurs in the phase when synaptic input from entorhinal cortex is strong and long-term potentiation (LTP) of excitatory connections arising from hippocampal region CA3 is strong, but synaptic currents arising from region CA3 input are weak”. Broadly speaking, this matches what we have here. That is to say: what drives CA1 during learning are not the synapses onto which learning is accumulating. Of course we don’t replicate this model in all its details – for example we don’t actually separate CA1 drive into two phases, and don’t model phase dependent LTD and so don’t reproduce their memory extinction results – but, philosophically, it is similar.

3) Lack of discussion of phase precession in CA1 cells. What are the theta firing patterns of CA1 (successor) cells in the model? Do they exhibit theta sequences and/or phase precession? We are never told this. The spike phase of the downstream CA1 cell is extremely important for STDP, as it determines whether synapses associated with past or future events are potentiated or suppressed (see Figure 8 of Chadwick et al. 2016, eLife). Based on my understanding, in the current setup CA1 place cells should produce phase precession during learning (before weights are updated), but only because each CA1 cell copies the activity of exactly one CA3 cell, which is unrealistic. Moreover, after the weights are updated, whether they produce phase precession is no longer clear. It is important to determine whether the proposed mechanism works in the more realistic scenario in which both CA3 and CA1 cells exhibit phase precession, but CA1 cells are driven by multiple CA3 cells.

Thank you for these suggestions. We now show in Figure 2—figure supplement 4 that the CA1 STDP successor features in the model do indeed inherit this phase precession:

The reason for this is that CA1 cells are still localised and therefore driven mostly by cells in CA3 which are close and which peak in activity together at a similar phase each theta cycle. As the agent moves through the CA1 cell it also moves through all the CA3 cells and their peak firing phase ‘precesses’ driving an earlier peak in the CA1 firing. Phase precession is CA1 after learning is noisier/broader than CA3 but far from non-existent and looks similar to real phase precession data from cells in CA1. This result is described in the main text:

“In particular, the parameters controlling phase precession in the CA3 basis features (Figure 2–supplement 4a) can affect the CA1 STDP successor features learnt, with ‘weak’ phase precession resembling learning in the absence of theta modulation (Figure 2–supplement 4bc), biologically plausible values providing the best match to the TD successor features (Figure 2–supplement 4d) and ‘exaggerated’ phase precession actually hindering learning (Figure 2–supplement 4e; see Supplementary Materials for more details). Additionally, we find these CA1 cells go on to inherit phase precession from the CA3 population even after learning when they are driven by multiple CA3 fields (Figure 2–supplement 4f).”

And we elaborate on this in the appendices/methods:

“Phase precession of CA1: In most results shown in this paper the weights are anchored to the identity during learning. This means each CA1 cells inherits phase precession from the one and only one CA3 cell it is driven by. It is important to establish whether CA1 still shows phase precession after learning when driven by multiple CA3 cells or, equivalently, during learning when the weights aren’t anchored and it is therefore driven by multiple CA3 neurons. Analysing the spiking data from CA1 cells after learning (phase precession turned on) shows it does phase precession. This phase precession is noisier than the phase precession of a cell in CA3 but only slightly and compares favourably to real phase precession data for CA1 neurons (panel f, right, with permission from Jeewajee et al. (2014) [46]).

The reason for this is that CA1 cells are still localised and therefore driven mostly by cells in CA3 which are close and which peak in activity together at a similar phase each theta cycle. As the agent moves through the CA1 cell it also moves through all the CA3 cells and their peak firing phase precesses driving an earlier peak in the CA1 firing. Phase precession is CA1 after learning is noisier/broader than CA3 but far from non-existent and looks similar to real phase precession data from cells in CA1.”

Additionally, by extending our parameter sweep to include phase precession parameters (Figure 2–supplement 3 panel c, last 2 subplots), we now show that the biologically derived values for the parameters determining the phase precession in the model are in fact optimally placed to approximate the TD learning of successor features (Figure 2–supplement 4, please see response to point 5 for more details).

Finally, we show that the CA1 successor features can still be successfully learnt via the STDP + phase precession mechanism when the target features are driven by multiple CA3 cells (Figure 2 supplement 2A), and when the target features are updated by the learnt weights online (Figure 2 supplement 2bc, please see response to point 2 for technical details).

4) Related to the preceding comment, there is a phase shift/delay between CA3 and CA1 (Mizuseki, Buzsaki et al., 2010). This doesn't seem to have been taken into account. Can the model be set up so that i) CA1 cells receive inputs from multiple CA3 cells ii) both CA3 and CA1 cells exhibit phase precession iii) there is the appropriate phase delay between CA3 and CA1?

Thank you for this comment, as it provoked much thought. At the level of individual cells in our model, the phase shift presented by Mizuseki, Buzsaki et al., 2010 (i.e. CA1 being shifted temporally just ahead of CA3 ) is functionally near-identical to if each CA3 basis feature were connected to a different CA1 cell slightly further ahead of it down the track. Therefore, in total, this would simply manifest as a rotation on the weight matrix (e.g. realignment of CA1 cells along the track). Thus perhaps these phase delays are important for other aspects of learning we are not capturing here. However, if this shift were more substantial, it is not entirely clear what would happen. We identify this as a limitation and direction for future work in the new paragraph we have added that discussing the limits of the model’s biological plausibility (reprinted below for convenience):

“While our model is biologically plausible in several respects, there remain a number of aspects of the biology that we do not interface with, such as different cell types, interneurons and membrane dynamics. Further, we do not consider anything beyond the most simple model of phase precession, which directly results in theta sweeps in lieu of them developing and synchronising across place cells over time [60]. Rather, our philosophy is to reconsider the most pressing issues with the standard model of predictive map learning in the context of hippocampus (e.g., the absence of dopaminergic error signals in CA1 and the inadequacy of synaptic plasticity timescales). We believe this minimalism is helpful, both for interpreting the results presented here and providing a foundation for further work to examine these biological intricacies, such as the possible effect of phase offsets in CA3, CA1 [61] and across the dorsoventral axis [62, 63], as well as whether the model’s theta sweeps can alternately represent future routes [64] by the inclusion of attractor dynamics [65].”

5) Dependence of learning on the noisiness of phase precession. The hyperparameter sweep seems to omit some of the most important variables, such as the spread paramaeter (kappa) and the place field width and running speed (see next comment). Since the successor representation is shown to be learned well when kappa=1 but not when kappa=0 (i.e. when phase precession is removed), this leaves open the question of what happens when kappa is bigger than or small than 1. It would be nice to see kappa systematically varied and the consequences explored.

Thank you for this suggestion. We have now extended our parameter sweep (Figure 2 supplement 3) to systematically determine the effect of variations in the noisiness of the phase precession (kappa) and the proportion of the theta cycle in which the precession takes place (β). Interestingly, we find that the biologically derived parameters are in fact optimally placed to approximate the TD learning of successor features (Figure 2 supplement 3c and 4a-e). We summarise these results in the main text:

“In particular, the parameters controlling phase precession in the CA3 basis features (Figure 2–supplement 4a) can affect the CA1 STDP successor features learnt, with ‘weak’ phase precession resembling learning in the absence of theta modulation (Figure 2–supplement 4bc), biologically plausible values providing the best match to the TD successor features (Figure 2–supplement 4d) and ‘exaggerated’ phase precession actually hindering learning (Figure 2–supplement 4e; see Supplementary Materials for more details). Additionally, we find these CA1 cells go on to inherit phase precession from the CA3 population (Figure 2–supplement 4f).”

In an additional supplementary figure (Figure 2–supplement 4) we delve into these hyperparameter sweep results showing examples of too-much or too-little phase precession on the learnt successor features and attempt to shed light on why this intermediate optima exist.

We also go into further detail in the appendices/methods:

“The optimality of biological phase precession parameters In figure 2 supplement 3 we ran a hyperparameter sweep over the two parameters associated with phase precession: κ, the von Mises parameter describing how noisy phase precession is and β, the fraction of the full 2π theta cycle phase precession crosses. The results show that for both of these parameters there is a clear “goldilocks” zone around the biologically fitted parameters we chose originally. When there is too much (large κ, large β) or too little (small κ, small β) phase precession performance is worse than at intermediate biological amounts of phase precession. Whilst – according to the central hypothesis of the paper – it makes sense that weak or non-existence phase precession hinders learning, it is initially counter intuitive that strong phase precession also hinders learning.

We speculate the reason is as follows, when β is too big phase precession spans the full range from 0 to 2π, this means it is possible for a cell firing very late in its receptive field to fire just before a cell a long distance behind it on the track firing very early in the cycle because 2π comes just before 0 on the unit circle. When κ is too big, phase precession is too clean and cells firing at opposite ends of the theta cycle will never be able to bind since their spikes will never fall within a 20 ms window of each other. We illustrate these ideas in figure 2 supplement 4 by first describing the phase precession model (panel a) then simulating spikes from 4 overlapping place cells (panel b) when phase precession is weak (panel c), intermediate/biological (panel d) and strong (panel e). We confirm these intuitions about why there exists a phase precession “goldilocks” zone by showing the weight matrix compared to the successor matrix (right hand side of panels c, d and e). Only in the intermediate case is there good similarity.”

6) Wide place fields and slow speeds. Place fields in the model have a diameter of 2 metres. This is quite big – bigger than typical place field sizes in the dorsal hippocampus (which often have around 30 cm diameter, or 15 cm radius). Moreover, the chosen velocity of 16 cm/s is quite slow, and rats often run much faster in experiments (30 cm/s and higher). With the chosen parameters, it takes the rodent 12.5 s to traverse a place field, which is unrealistically long. My concern is that this setup leads to a large number of spikes per pass through a place field and that this unrealistic setting is needed for the proposed mechanism to learn effectively in a reasonable number of laps. What happens when place fields are smaller and running speeds faster, as is typically found in experiments? How many laps are required for convergence?

Thank you for this suggestion, we now explore this in a new fsupplementary figure, (Figure 2–supplement 1bc). In summary, we find there is no critical effect on learning with smaller place fields and faster speeds. As hypothesised by the reviewer, we find that the learning is slower (when measured in number of laps) due to the decreased number of spikes, but not with catastrophic effects. This is summarised in the results:

“Thus, the ability to approximate TD learning appears specific to the combination of STDP and phase precession. Indeed, there are deep theoretical connections linking the two – see Methods section 5.8 for a theoretical investigation into the connections between TD learning and STDP learning augmented with phase precession. This effect is robust to variations in running speed (Figure 2–supplement 1b) and field sizes (Figure 2–supplement 1c), as well as scenarios where target CA1 cells have multiple firing fields (Figure 2–supplement 2a) that are updated online during learning (Figure 2–supplement 2bc; see Supplementary Materials for more details)”

And elaborated on in the appendices/methods:

“Smaller place cells and faster movement: Nothing fundamental prevents learning from working in the case of smaller place fields or faster movement speeds. We explore this in figure 2 supplement 1, panel c, as follows: the agent speed is doubled from 16 cm s^−1^ to 32 cm s^−1^ and the place field size is shrunk by a factor of 5 from 2 m diameter to 40 cm diameter. To facilitate learning we also increase the cell density along the track from 10 cells m^−1^ to 50 cells m^−1^. We also shrink the track size from 5 m to 2 m (any additional track is redundant due to the circular symmetry of the set-up and small size of the place cells). We then train for 12 minutes. This time was chosen since 12 minutes moving at 32 cm s^−1^ on a 2 m track means the same number of laps as 60 mins moving at 16 cm s^−1^ on a 5 m track (96 laps in total). Despite these changes the weight matrix converged with high similarity to the successor matrix with a shorter time horizon (0.5 s). Convergence time measured in minutes was faster than in the original case but this is mostly due to the shortened track length and increased speed. Measured in laps it now takes longer to converge due to the decreased number of spikes (smaller place fields and faster movement through the place fields). This can be seen in the shallower convergence curve, panel c (right) relative to panel a.”

7) Running speed-dependence of phase precession and firing rate. The rat is assumed to run at a fixed speed – what happens when speed is allowed to vary? Running speed has profound effects on the firing of place cells, including i) a change in their rate of phase precession ii) a change in their firing rate (Huxter et al., 2003). More simulations are needed in which running speed varies lap-by-lap, and/or within laps.

Thank you for this suggestion, we now explore this in a new supplementary figure, (Figure 2–supplement 1b, see comment above) where the speed of the rat / agent is allowed to vary smoothly and stochastically. In summary, we find no observable effect on the STDP weight matrix or the TD successor matrix after learning, with the R^2 value between the two. This is summarised in the results:

“Thus, the ability to approximate TD learning appears specific to the combination of STDP and phase precession. Indeed, there are deep theoretical connections linking the two – see Methods section 5.8 for a theoretical investigation into the connections between TD learning and STDP learning augmented with phase precession. This effect is robust to variations in running speed (Figure 2–supplement 1b) and field sizes (Figure 2–supplement 1c), as well as scenarios where target CA1 cells have multiple firing fields (Figure 2–supplement 2a) that are updated online during learning (Figure 2–supplement 2bc; see Supplementary Materials for more details)”

With further details in the appendices/methods:

“Movement speed variability: Panel b shows an experiment where we reran the simulation shown in paper figures 2a-e except, instead of a constant motion speed, the agent moves with a variable speed drawn from a continuous stochastic process (an Ornstein-Uhlenbeck process). The parameters of the process were selected so the mean velocity remained the same (16 cm s^−1^ left-to-right) but now with significant variability (standard deviation of 16 cm s^−1^ thresholded so the speed can’t go negative). Essentially, the velocity takes a constrained random walk. This detail is important: the velocity is not drawn randomly on each time step since these changes would rapidly average out with small dt, rather the change in the velocity (the acceleration) is random – this drives slow stochasticity in the velocity where there are extended periods of fast motion and extended periods of slow motion. After learning there is no substantial difference in the learned weight matrices. This is because both TD and STDP learning rules are able to average-over the stochasticity in the velocity and converge on representations representative of the mean statistics of the motion.”

8) Two-dimensional phase precession. There is debate over how 2D environments are encoded in the theta phase (Chadwick et al. 2015, 2016; Huxter et al., 2008; Climer et al., 2013; Jeewajee et al., 2013). This should be mentioned and discussed – how much do the results depend on the specific assumptions regarding phase precession in 2D? For example, Huxter et al. found that, when animals pass through the edge of a place field, the cell initially precesses but then processes back to its initial phase, but this isn't captured by the model used in the present study. Chadwick et al. (2016) proposed a model of two-dimensional phase precession based on the phase locking of an oscillator, which reproduces the findings of Huxter et al. and makes different predictions for phase precession in two dimensions than the Jeewajee model used by the authors. It would be nice to test alternative models for 2D phase precession and determine how well they perform in terms of generating successor-like representations.

Thank you for this suggestion. We agree this is an important topic in terms of understanding the correlates and consequences of phase precession. There is a wealth of literature surrounding this topic, some of which we relied upon for defining the model of 2D phase precession implemented here (e.g. Jeewajee et al., 2013 and Chadwick et al. 2015). However, we believe that this would be better suited as a followup to the current study, which addresses the first question of what how closely the representations learned with classical theta precession resemble TD-trained SRs. Rather, we agree that considering alternative 2D models of phase precession would be a wonderful direction for future work and our code is publicly available should anyone wish to explore this.

9) Modelling the distribution of place field sizes along the dorsoventral axis. Two important phenomena were omitted that are likely important and could alter the conclusions. First, there is a phase gradient along the dorsoventral axis, which generates travelling theta waves (Patel, Buszaki et al., 2012; Lebunov and Siapas, 2009). How do the results change when including a 180 (or 360) phase gradient along the DV axis? The authors state that "A consequence of theta phase precession is that the cell with the smaller field will phase precess faster through the theta cycle than the other cell – initially it will fire later in the theta cycle than the cell with a larger field, but as the animal moves towards the end of the small basis field it will fire earlier" – this neglects to consider the phase gradient along the DV axis (see also Leibold and Monsalve-Mecado, 2017). Second, the authors chose three discrete place field sizes for their dorsoventral simulations. How would these simulations look if a continuum of sizes were used reflecting the gradient along the dorsoventral axis? Going further, CA1 cells likely receive input from CA3 cells with a distribution of place field sizes rather than a single place field size – how would the model behave in that case?

Thank you for this interesting point. The model and results presented here pertain more to the role of theta compression (and STDP) in approximating TD learning. However we have now added the following to our discussion to consider these additional aspects of theta oscillations:

“The distribution of place cell receptive field size in hippocampus is not homogeneous. Instead, place field size grows smoothly along the longitudinal axis (from very small in dorsal regions to very large in ventral regions). Why this is the case is not clear – our model contributes by showing that, without this ordering, large and small place cells would all bind via STDP, essentially overwriting the short timescale successor representations learnt by small place cells with long timescale successor representations. Topographically organising place cells by size anatomically segregates place cells with fields of different sizes, preserving the multiscale successor representations. The functional separation of these spatial scales could be further enhanced by a gradient of phase offsets along the dorso-ventral axis, resulting from the theta oscillation being a travelling wave [62, 63]. This may act as a temporal segregation preventing learning between cells of different field sizes, on top of the anatomical segregation we explore here. The premise that such separation is needed to learn multiscale successor representations is compatible with other theoretical accounts for this ordering. Specifically Momennejad and Howard [39] showed that exploiting multiscale successor representations downstream, in order to recover information which is ‘lost’ in the process of compiling state transitions into a single successor representation, typically requires calculating the derivative of the successor representation with respect to the discount parameter. This derivative calculation is significantly easier if the cells – and therefore the successor representations – are ordered smoothly along the hippocampal axis.”

As well as this, we include a new paragraph in the discussion pertaining to these limits in the model’s biological plausibility and our intended contribution:

“While the model is biologically plausible in several respects, there remain a number of aspects of the biology that we do not interface with, such as different cell types, interneurons and membrane dynamics. Further, only the most simple model of phase precession is considered, which directly results in theta sweeps in lieu of them developing and synchronising across place cells over time [60]. Rather, our philosophy is to reconsider the most pressing issues with the standard model of predictive map learning in the context of hippocampus. These include the absence of dopaminergic error signals in CA1 and the inadequacy of synaptic plasticity timescales. We believe this minimalism is helpful, both for interpreting the results presented here and providing a foundation on which further work may examine these biological intricacies, such as the possible effect of phase offsets in CA3, CA1 [61] and across the dorsoventral axis [62, 63], as well as whether the model’s theta sweeps can alternately represent future routes [64] e.g. by the inclusion of attractor dynamics [65].”

– There is no theoretical analysis of why theta sequences+STDP approximates the TD algorithm, or when the proposed mechanism might/might not work. The model is simple enough that some analysis should be possible. It would be nice to see this elaborated on – can a reduced model be obtained that captures the learning algorithm embodied by theta sequences+STDP, and does this reduced model reveal an explicit link to the TD algorithm? If not, then why does it work, and when might it generalise/not work?

Thank you for this suggestion. We have now updated the manuscript to include a section (Methods 5.8) explaining the theoretical connection between STDP and TD learning. In short, it starts by showing how temporal difference learning can be mathematically recast into a temporally asymmetric Hebbian learning rule reminiscent of simplified STDP. However, in order to recast TD learning in its STDP-like form it is necessary to fix the temporal discount time horizon to the synaptic plasticity timescale. This alone would produce TD-style learning on a time-scale too short to capture meaningful predictions of behaviour. Thus, we show mathematically that the importance of theta phase precession is to provide a precise temporal compression on the input sequences that effectively increases this predictive time horizon from the timescale of synaptic plasticity to the timescale of behaviour. This temporal compression overcomes the timescales problem since, by symmetry, learning a successor feature with a very small time horizon where the input trajectory is temporally compressed is equivalent to learning a successor feature with a long time horizon where the inputs are not compressed. We derive a formula for the amount of compression as a function of the typical speed of a `theta sweep’ and estimate a ballpark figure showing that in many cases this compression is enough to extend the synaptic plasticity timescale into behaviourally relevant timescales. In essence, this section provides the mathematics behind the very intuition on which we based the study (e.g. Figure 1). That is:

1. Fundamentally, STDP behaves similarly to TD learning since the temporally asymmetric learning rule binds pairs of cells if one cell spikes before (i.e. is predictive of) the other.

2. STDP can’t easily learn temporally extended predictive maps but can if phase precession “compresses” input features.

Finally, we end this theoretical analysis section by examining where and why the two learning rules diverge (i.e. where STDP does not approximate TD learning). We direct the reader to studies that focus more closely on modified Hebbian learning rules to circumvent these issues, whilst pointing out that it does not have to be one or the other – the intuition for why theta phase precession helps learning applies equally well to modified learning rules which focus more closely on exactly replicating TD learning at the expense of similarity to biological STDP. We include the newly added theory section at the end of this review response document.

– The comparison of successor features to neural data was qualitative rather than quantitative, and often quite vague. This makes it hard to know whether the predictions of the model are actually consistent with real neural data. It would be much preferred if a direct quantitative comparison of the learned successor features to real data could be performed, for example, the properties of place fields near to doorways.

We agree that we could be much more specific in our comparisons to neural data, and that making quantitative comparisons to experimental recorded place cells would be a valuable contribution. To address the first point, we have clarified the presentation of our results in several places in order to make the connections to existing neural data more specific. As for making comparisons to data, we believe it is outside the scope of this work. Our primary contribution is to make quantitative comparisons between successor representations learned by TD and learned by STDP+theta. This led us to testable predictions that we have described in the discussion (page 11, paragraph beginning “Our theory makes the prediction”) that specifically relate to the effect of impairing theta oscillations at different stages of learning (we note that these descriptions have been rewritten to be clearer in the revised manuscript). We believe that these kind of experiments would be optimal for providing datasets that would be better suited for the specific theoretical questions we are investigating here than would a post-hoc analysis of an existing datasets. Finally, we have now included theoretical analysis of the connection between STDP and TD learning (see comment above), in which readers may find a more insightful way to gain intuition about how closely this model matches SR theory and solidifies the theory contribution.

We also want to note that some prior (and in-review) work has conducted quantitative comparisons between hippocampal data and successor representations. Neuroimaging studies have shown evidence for predictive coding of spatial and non-spatial states on varying time-horizons (Garvert et al. 2017, Schapiro et al. 2016, Brunec and Momennejad 2022). Other studies have found that the SR did not explain under certain conditions, such as Duvelle et al. 2021. de Cothi et al. 2022 provide a model comparison to explain navigation behaviours in humans and rats, and found that both were best explained by a successor representation-like strategy. We also note that in recent work also under review at *eLife*, Ching Fang and colleagues conduct a quantitative comparison between place fields recorded from chickadees and the successor representation (Fang et al. 2022).

– Statistical structure of theta sequences. The model used by the authors is identical to that of Chadwick et al. (2015) (except for the thresholding of the Gaussian field), and so implicitly assumes that theta sequences are generated by the independent phase precession of each place cell. However, the authors mention in the introduction that other studies argue for the coordination of place cells, such that theta sequences can represent alternative futures on consecutive theta cycles (Kay et al.). This begs the question: how important is the choice of an independent phase precession model for the results of this study? For example, if the authors were to simulate a T-maze, would a model which includes cycling of alternative futures learn the successor representation better or worse than the model based on independent coding? Given that there now is a large literature exploring the coordination of theta sequences and their encoded trajectories, it would be nice to see some discussion of how the proposed mechanism depends on/relates to this.

Thank you for this suggestion. We have added a citation to Chadwick *et al.*, 2015 (ref [42]) as well as the following at the beginning of the results:

“As the agent traverses the receptive field, its rate of spiking is subject to phase precession fjθ(x,t) with respect to a 10 Hz theta oscillation. This is implemented by modulating the firing rate by an independent phase precession factor which varies according to the current theta phase and how far through the receptive field the agent has travelled [42] (see Methods and Figure 1a)”

We also discuss limits of the model with regard to the Kay et al. study, as well as possible manipulations to capture this result, in a new discussion paragraph:

““While our model is biologically plausible in several respects, there remain a number of aspects of the biology that we do not interface with, such as different cell types, interneurons and membrane dynamics. Further, we do not consider anything beyond the most simple model of phase precession, which directly results in theta sweeps in lieu of them developing and synchronising across place cells over time [60]. Rather, our philosophy is to reconsider the most pressing issues with the standard model of predictive map learning in the context of hippocampus (e.g., the absence of dopaminergic error signals in CA1 and the inadequacy of synaptic plasticity timescales). We believe this minimalism is helpful, both for interpreting the results presented here and providing a foundation for further work to examine these biological intricacies, such as the possible effect of phase offsets in CA3, CA1 [61] and across the dorsoventral axis [62, 63], as well as whether the model’s theta sweeps can alternately represent future routes [64] by the inclusion of attractor dynamics [65].”

And elaborate on both of these points in the methods section:

“Our phase precession model is “independent” (essentially identical to Chadwick et al. (2015)[42]) in the sense that each place cell phase precesses independently from what the other place cells are doing. In this model, phase precession directly leads to theta sweeps as shown in Figure 1. Another class of models referred to as “coordinated assembly” models [76] hypothesise that internal dynamics drive theta sweeps within each cycle because assemblies (aka place cells) dynamically excite one-another in a temporal chain. In these models theta sweeps directly lead to phase precession. Feng and colleagues draw a distinction between theta precession and theta sequence, observing that while independent theta precession is evident right away in novel environments, longer and more stereotyped theta sequences develop over time [77]. Since we are considering the effect of theta precession on the formation of place field shape, the independent model is appropriate for this setting. We believe that considering how our model might relate to the formation of theta sequences or what implications theta sequences have for this model is an exciting direction for future work.”

[Editors' note: further revisions were suggested prior to acceptance, as described below.]

The manuscript has been improved but there are some remaining issues that need to be addressed, as outlined below:1. Spiking model. We all agree with you that a full spiking model would be much too complex. However, since you already generate spikes using a Poisson process, it would be useful to see a simulation where the Poisson rate of CA1 cell is determined by the integration of the incoming CA3 spikes (perhaps with many incoming CA3 neurons). If this doesn't work, you should discuss why this is the case and what the implications are for the model.

Thank you for this suggestion. In order to address points 1 and 2 (see below), we have updated the manuscript to include a new simulation where the spiking activity in CA1 cells (N=50) is driven by a large number of spiking CA3 neurons (N=500) with overlapping fields that phase precess. To avoid the complexity of Hodgkin-Huxley / Leaky integrate-and-fire models, spiking activity in CA1 is determined by a linear-nonlinear cascade model, which we would like to thank Reviewer 2 for suggesting. In the simulation, which has been added as a panel in Figure 2 Supplement 2, we find that the resulting weights learnt via STDP in the spiking model are almost identical to those learnt by the standard STDP-successor learning rule used in most of our previous simulations (diagonal-aligned average across rows: R^2^=0.99). Note that the resulting weight matrix is no longer square due to the x10 greater number of cells in CA3 vs. CA1.

The results of the simulation are also now referred to in the Results:

“This effect is robust to variations in running speed (Figure 2–supplement 1b) and field sizes (Figure 2–supplement 1c), as well as scenarios where target CA1 cells have multiple firing fields (Figure 2–supplement 2a) that are updated online during learning (Figure 2–supplement 2b-d), or fully-driven by spikes in CA3 (Figure 2–supplement 2e); see methods for more details.”

and details of the simulations have been added to the Methods section 5.10.2 (equations for the spiking model are also summarised in the figure above).

2. CA3 => CA1 projections. CA1 cells still receive input from just one CA3 cell for each place field in the updated model (at least in the majority of simulations). This allows precise theta timing of the pre and post -synaptic neurons which appears to be critical for the plasticity rule to function. For example, the mathematics of Geisler et al. 2007 shows that, if the CA1 cell would receive input from a set of phase precessing CA3 cells with spatially offset place field and a Gaussian weight profile (the most common way to model CA3-CA1 connections), then the CA1 cell would actually fire at the LFP theta frequency and wouldn't phase precess, and as a consequence the STDP mechanism would no longer learn the successor representation. This suggests strong constraints on the conditions under which the model can function which are currently not being adequately discussed. This should be investigated and discussed, and the constraints required for the model to function should be plainly laid out.

We agree that the one-to-one nature between CA3 => CA1 in the majority of simulations might suggest that this is a strict condition in order for the model to function. Rather, it is a condition we impose in order to simplify the model as well as to establish a clear connection between the model and successor feature theory (see Methods section 5.9). However it is important to note that this condition can easily be relaxed and that doing so still produces results that are extremely similar to true successor feature learning. In order to show this, we have updated the manuscript to include a new simulation, also outlined above, where the spiking activity in CA1 cells (N=50) is driven by a large number of spiking CA3 neurons (N=500) with overlapping fields that phase precess. We show that in this regime, even when the spiking output of each CA1 cell is determined by the spiking input of a large number of phase precessing CA3 cells, the resulting STDP synaptic weights closely resembles that of the STDP-successor matrix used in the majority of simulations (diagonal-aligned average across rows: R^2^=0.99).

Additionally we also show a similar result in Figure 2 supplement 2bandc (added after the first round of review) where the synaptic weight matrix is updated online, during learning. In these models all 50 (not just one) CA3 cells are able to drive CA1 cells during learning and SR-like weight matrices still develop. In total, we believe these simulations and the new one performed here demonstrate that many-to-one projections do not pose a fundamental issue.

Regarding the effect on phase precession (or the lack thereof) when cells are driven by multiple neurons, the simulation described above, and the ones provided in response to the first round of reviews, show that this is not a substantial concern. Geisler et al. (2010) raised the possibility that in such a situation phase precession would not emerge in CA1. However, our simulations show that it is possible for CA1 cells receiving input from *multiple* CA3 cells to phase precess as long as there is some spatial structure to the connections. If a CA1 cell is most strongly driven by a population of CA3 cells in a similar location on the track (and which therefore phase precess similarly) it too will phase precess. This spatial structure can be quite broad, for evidence of this please see Figure 2 supplement 2f included in our previous rebuttal, and Author response image 1 for an equivalent plot drawn from the spiking model simulation described above. In the figures we show the phase precession of CA1 when driven by the learnt synaptic weight matrix, W, which is significant over a large portion of the input CA3 neurons. This demonstrates that many-to-one connections are not incompatible with phase precession and therefore our proposed learning mechanism can still work.

**Author response image 1. sa2fig1:** In a fully spiking model, CA1 neurons inherit phase precession from multiple upstream CA3 neurons. Top, model schematic – each CA1 neuron receives input from multiple CA3 neurons with contiguous place fields. Bottom, position vs phase plot for an indicative CA1 neuron, showing strong phase precession similar to that observed in the brain.

3. A similar concern holds with the phase offset between CA3 and CA1 found by Mizuseki et al. The theta+STDP mechanism learns the successor representation because the CA1 cells inherit their responses from a phase-precessing upstream CA3 cell, so the existence of a phase lag is troubling, because it suggests that CA1 cells are not driven causally by CA1 cells in the way the model requires. You may be right that, if some external force were to artificially impose a fixed lag between the CA3 and CA1 cell, the proposed learning mechanism would still function but now with a spatial offset. However, the Reviewer was concerned that the very existence of the phase lag challenges the basic spirit of the model, since CA1 cells are not driven by CA3 cells in the way that is required to learn causal relationships. At the very least, this needs to be addressed and discussed directly and openly in the Discussion section, but it would be better if the authors could implement a solution to the problem to show that the model can work when an additional mechanism is introduced to produce the phase lag (for example, a combination of EC and CA3 inputs at different theta phases?)

The reviewer is correct in that since there is a theta phase offset between CA3 and CA1, it is important to consider the possible impact on our model. Indeed, while Mizuseki et al., 2009 alludes to a fixed phase difference between CA3 and CA1 neurons, the consequences for phase precessing place cells are more nuanced. Importantly, in a later paper from 2012, Misuzeki et al. demonstrate this offset in phase between CA3 and CA1 place cells varies at different stages of the theta cycle. Thus as an animal first enters a place field and spikes are fired late in the theta cycle, CA1 spikes are emitted around 80°to 90° after spikes from CA3. However, as the animal progresses through the field, spikes from both regions precess to earlier phases but the effect is more pronounced in CA1, meaning that by the time the animal exits a place fields the phase offset between the two regions is essentially 0° (the key figure from Mizuseki et al. 2012 is shown in Figure 2 Supplement 4g). Importantly this result fits with the work of Hasselmo et al. (2002) and Colgin et al. (2009) both of which point to there being enhanced CA3 > CA1 coupling at early theta phases – in other words CA3’s influence on CA1 appears to be most pronounced in the latter half of place fields.

In response to this we have done two things. First, to simulate the effect of a variable phase offset, we ran the model as before but for offsets of 90°, 45°, and 0°, which correspond to late, mid and early theta phase. We then averaged the resulting STDP weight matrices to generate a single prediction for a system in which the CA3 to CA1 phase offset varies in a plausible fashion – the resulting matrix is still very similar to the TD successor matrix (diagonal-aligned average across rows: R^2^=0.76), and clearly shows the SR-like asymmetry (positive band left of diagonal, negative band right) confirming that our model is robust to the observed phase offset. These simulations, including the weight matrices for offsets of 90°, 45°, and 0° have now been included in a new figure panel appended to Figure 2 Supplement 4:

and are referred to in the Results section:

“Additionally, we find these CA1 cells go on to inherit phase precession from the CA3 population even after learning when they are driven by multiple CA3 fields (Figure 2–supplement 4f), and that this learning is robust to realistic phase offsets between the populations of CA3 and CA1 place cells (Figure 2—figure supplement 4g).”

Secondly, we have also updated the discussion to cover these points in more detail and in particular have addressed the nuances suggested by the experimental results from Hasselmo et al. (2002) and Colgin et al. (2009). Specifically, we indicate that because CA3>CA1 coupling is most pronounced at early theta phases – when the phase offset between the regions is at its lowest – the effect of the offset is likely to be less important than might immediately be thought. Thus the simulation presented above, which still learns a good approximation of the TD SR matrix (diagonal-aligned average across rows: R^2^=0.76), should be considered as a worst-case scenario.

We now expand upon these points in the Discussion:

“While the model is biologically plausible in several respects, there remain a number of aspects of the biology that we do not interface with, such as different cell types, interneurons and membrane dynamics. Further, we do not consider anything beyond the most simple model of phase precession, which directly results in theta sweeps in lieu of them developing and synchronising across place cells over time [60]. Rather, our philosophy is to reconsider the most pressing issues with the standard model of predictive map learning in the context of hippocampus (e.g., the absence of dopaminergic error signals in CA1 and the inadequacy of synaptic plasticity timescales). We believe this minimalism is helpful, both for interpreting the results presented here and providing a foundation on which further work may examine these biological intricacies, such as whether the model’s theta sweeps can alternately represent future routes [61] e.g. by the inclusion of attractor dynamics [62]. Still, we show this simple model is robust to the observed variation in phase offsets between phase precessing CA3 and CA1 place cells across different stages of the theta cycle [63]. In particular, this phase offset is most pronounced as animals enter a field (∼90°) and is almost completely reduced by the time they leave it (~0°) (Figure 2 —figure supplement 4g). Essentially our model hypothesises that the majority of plasticity induced by STDP and theta phase precession will take place in the latter part of place fields, equating to earlier theta phases. Notably, this is in-keeping with experimental data showing enhanced coupling between CA3 and CA1 in these early theta phases [64, 65]. However, as our simulations show (figure 2 supplement 4 panel g ), even if these assumptions do not hold true, the model is sufficiently robust to generate SR equivalent weight matrices for a range of possible phase offsets between CA3 and CA1 ”

with details of the simulations added to the Methods:

“Phase shift between CA3 and CA1. In figure 2 supplement 4g we simulate the effect of a decreasing phase shift between CA3 and CA1. As observed by Mizuseki et al. (2012) [87] there is a phase shift between CA3 and CA1 neurons being maximally around 90 degrees at the end of each theta cycle, decreasing to 0 at the start. We simulate this by adding a temporal delay to all downstream CA1 spikes equivalent to the phase shifts of 0°, 45° and 90°. The average of the weight matrices learned over all three examples still displays clear SR-like structure.”

4. DV phase precession. The Reviewer would still like to see you introduce DV phase lags, which could be done with a simple modification of the existing simulations. At minimum, it is critical to remove/modify the sentence "A consequence of theta phase precession is that the cell with the smaller field will phase precess faster through the theta cycle than the other cell – initially it will fire later in the theta cycle than the cell with a larger field, butas the animal moves towards the end of the small basis field it will fire earlier." As R2 noted in their original review, this is not the case when DV phase lags are taken into account, as was shown by Leibold and Monsalve-Mercado (2017). Ideally, it would be best to update simulations updated to account for the DV phase lags and the discussion updated to account for their functional implicationsReviewer #2 (Recommendations for the authors):While the reviewers have undertaken a number important additional analyses which address some of the concerns raised in the review, several of the most pressing concerns regarding biological plausibility have not been addressed. In particular, each CA1 place field is still inherited by exactly 1 CA3 place field in the updated protocol, and cells still interact via their firing rates with spikes only being used for the weight updates. Moreover, the authors chose not to address concerns regarding quantitative comparisons between the model and data. Overall, while the authors correctly point out that their primary contribution should be viewed as illustrating a mechanism to learn successor representations via phase precession and STDP, this message is undermined if the proposed mechanism can't function when reasonable assumptions are made regarding the number of cells and their mode of interaction.

Thank you for highlighting this. The sentence mentioned was actually intended to be a ‘strawman’ to motivate the subsequent analyses that show the different rates of phase precession induced by varied field sizes do not impair plasticity in a manner that is sufficient to segregate spatial scales (Figure 4). Note, to be clear we were referring to the fact that small place fields – found at the dorsal pole of the hippocampus – do phase precess more rapidly in *time.* The point being that phase precession is proportional to field size, so for a given distance travelled – say 20cm – a small field will exhibit a greater change in spiking phase than a large one. We apologise for presenting this information in a way that was not clear. We have now made the following changes to ensure the intention of this paragraph is clear:

“Hypothetically, consider a small basis feature cell with a receptive field entirely encompassed by that of a larger basis cell with no theta phase offset between the entry points of both fields. A potential consequence of theta phase precession is that the cell with the smaller field would phase precess faster through the theta cycle than the other cell – initially it would fire later in the theta cycle than the cell with a larger field, but as the animal moves towards the end of the small basis field it would fire earlier. These periods of potentiation and depression instigated by STDP could act against each other, and the extent to which they cancel each other out would depend on the relative placement of the two fields, their size difference, and the parameters of the learning rule.”

Similarly, as outlined in the simulations above, graduated theta phase offsets of up to and including 90° are also insufficient to impair the plasticity induced by STDP and phase precession. Applying both of these findings in the context of theta as a travelling wave across the dorsal-ventral axis, our original conclusion that topographic organisation of place cells by size along the DV axis is necessary to prevent cross binding and preserve multiscale structure in the resulting successor features remains unchanged.

We now clarify these points in the discussion:

“The distribution of place cell receptive field size in hippocampus is not homogeneous. Instead, place field size grows smoothly along the longitudinal axis (from very small in dorsal regions to very large in ventral regions). Why this is the case is not clear – our model contributes by showing that, without this ordering, large and small place cells would all bind via STDP, essentially overwriting the short timescale successor representations learnt by small place cells with long timescale successor representations. Topographically organising place cells by size anatomically segregates place cells with fields of different sizes, preserving the multiscale successor representations. Further, our results exploring the effect of different phase offsets on STDP-successor learning (Figure 2 —figure supplement 4g) suggest that the gradient of phase offsets observed along the dorso-ventral axis [79, 80] is insufficient to impair the plasticity induced by STDP and phase precession. The premise that such separation is needed to learn multiscale successor representations is compatible with other theoretical accounts for this ordering. Specifically Momennejad and Howard [39] showed that exploiting multiscale successor representations downstream, in order to recover information which is ‘lost’ in the process of compiling state transitions into a single successor representation, typically requires calculating the derivative of the successor representation with respect to the discount parameter. This derivative calculation is significantly easier if the cells – and therefore the successor representations – are ordered smoothly along the hippocampal axis.”